

# The role of debris cover in the evolution of Zmuttgletscher, Switzerland, since the end of the Little Ice Age

Nico Mölg[1], Tobias Bolch[1], Andrea Walter[1], Andreas Vieli[1]

[1]Department of Geography, University of Zurich, Zurich, 8057, Switzerland

*Correspondence to*: Nico Mölg (nico.moelg@geo.uzh.ch)

**Abstract.** Debris-covered glaciers often exhibit large, flat, slow-flowing tongues. Many of these glaciers show high thinning rates today despite thick debris cover. Due to lack of observations, most existing studies have neglected the dynamic interaction between debris cover and glacier evolution over longer time periods. The main aim of this study is to reveal this interaction by reconstructing changes of debris cover, glacier geometry, flow velocities, and surface features of Zmuttgletscher (Switzerland),
based on historic maps, satellite images, aerial photographs, and field observations. We show that debris cover extent has increased from ~13% to >32% of the total glacier surface since 1859 and that the debris is sufficiently thick to reduce ablation compared to bare ice over much of the ablation area. Despite the debris cover the volume loss of Zmuttgletscher is comparable to that of debris-free glaciers located in similar settings whereas changes in length and area have been small in comparison. Increased ice mass input in the 1970s and 1980s resulted in a temporary velocity increase, as well as a lowering of the upper
margin of debris cover and exposed-ice area, and a reduction of ice cliffs. Since ~2001, the lowest ~1.5 km are stagnant despite a slight increase in surface slope of the glacier tongue. We conclude that the debris cover governs the pattern of volume loss without changing its magnitude, which is due to the large ablation area and strong thinning in regions with thin debris further up-glacier and in the regions of meltwater channels and ice cliffs. At the same time rising temperatures lead to increasing debris cover and decreasing glacier dynamics, thereby slowing down length and area losses.

## 1   Motivation and Objectives

Debris-covered glaciers have been observed to show a delayed adjustment of their length to climatic changes (e.g. Ogilvie, 1904; Scherler et al., 2011). This behaviour can be explained by melt reduction due to insulation by the debris layer, which commonly increases in thickness towards the terminus (Östrem, 1959; Nakawo et al., 1986; Anderson and Anderson, 2018), and is expected to distinctly prolong the glacier's response time (Jóhannesson et al., 1989). In some regions debris-covered
glaciers exhibit similar rates of volume changes as debris-free glaciers (e.g. Kääb et al., 2012; Gardelle et al., 2012; Nuimura et al., 2012). Proposed reasons for this behaviour are: the emergence of surface features with exposed ice (ice cliffs, water flow channels, and ponds), stronger thinning further up-glacier that compensates for the debris-induced melt reduction, reduced mass flux from the accumulation areas and decreasing emergence velocities at the tongue, and englacial ablation with subsequent collapsing of the glacier surface (Ragettli et al., 2016; Banerjee, 2017). Ice cliffs and ponds can enhance ablation
in comparison to debris-covered surfaces and even debris-free ice and are common features on debris-covered glacier tongues (Brun et al., 2016; Benn et al., 2012). Especially during periods of negative mass balance, the down-glacier increase of debris-cover thickness reduces ablation through its insulating effect and can lead to a lower – and even reversed – mass balance gradient (Nakawo et al., 1999; Benn and Lehmkuhl, 2000; Benn et al., 2012; Ragettli et al., 2016; Rounce et al., 2018). Over time, the surface slope of the glacier tongue therefore often reduces, which can lead to a decrease in driving stress and thus ice
flow velocity (Kääb, 2005; Bolch et al., 2008b; Quincey et al., 2009; Jouvet et al., 2011; Rowan et al., 2015). Furthermore, with an increase in equilibrium line altitude (ELA) the englacial debris melts out faster, leading to an extended debris cover further up-glacier (Kirkbride and Deline, 2013; Carturan et al., 2013). As a result, strongly debris-covered glaciers often have long, flat, and low-lying tongues with low flow velocities or even stagnant parts at their tongues (Benn et al., 2012). Current



research investigates the involved processes, i.e. ablation under a debris cover and in areas of ice cliffs and ponds, ablation and thinning at the glacier tongue, changes of debris cover over time, or surface flow velocities, at many debris-covered glaciers, specifically in the Himalaya (e.g. Hambrey et al., 2008; Bolch et al., 2012; Benn et al., 2012; Dobhal et al., 2013; Pellicciotti et al., 2015; Gibson et al., 2017); some studies have started to use the existing knowledge about the mutual influence

of some of these processes to numerically model the glaciers' development (Jouvet et al., 2011; Rowan et al., 2015; Anderson and Anderson, 2016; Banerjee, 2017).

However, most studies face difficulties leading to persistent shortcomings in our understanding of the development of debris-covered glaciers: time series are often short and with a few decades well below the expected response time of larger glaciers; investigations are often local because repeated tongue-wide data are sparse, especially for longer periods; only one or few

variables are considered, and thus the mutual influences of e.g. changing debris cover and glacier geometry cannot be assessed; studies at glacier-scale over more than a decade have mostly been conducted in the Himalaya (e.g. Bolch et al., 2008b; Ragettli et al., 2016; Lamsal et al., 2017).

To better understand how a changing debris cover affects glacier geometry, flow velocities, and surface features, and how it is in turn affected by these variables, it is necessary to consider the development of glaciers beyond their potential response times.

Few studies have investigated the temporal evolution of debris cover on glaciers (overview in Kirkbride and Deline, 2013), or the evolution of debris-covered glaciers over time (e.g. Agata & Zanutta 2007; Capt et al., 2016). Several studies observed an increase of debris cover on glaciers during negative mass balance periods (Kirkbride and Deline, 2013; Quincey and Glasser, 2009; Bolch et al., 2008a). Because of the overall negative mass balance trend, glaciers are and will be increasingly affected by debris cover. It is important to understand how this increase affects the geometry and dynamics of the glaciers in order to

simulate their future development (Jouvet et al., 2011; Anderson and Anderson, 2016).

Learning about the long-term effects of debris cover will be valid not only for glaciers in the Alps, but for glaciers worldwide that undergo general retreat. Large glaciers in the Himalaya and Karakoram – which are in the centre of attention due to their importance – are not ideal candidates for long-term investigations due to long response times (>50 years) and data scarcity before the 1960s. In contrast, the long history of length monitoring (GLAMOS, 2018) as well as the availability of topographic

maps and aerial photos from the mid-19[th] century onwards make Swiss glaciers well suited sites to study long-term developments; the earliest maps (1850s, 1870s) already include debris symbols and contour lines, and stereo imagery throughout the 20[th] century allows for detailed 3D surface investigations.

In this study we aim to understand the long-term geometric evolution and dynamics of debris-covered glaciers at the example of Zmuttgletscher in the Swiss Alps. This medium-sized valley glacier has been going through the transition from a mostly

debris-free glacier in the late 1850s to one that is almost completely debris-covered in its ablation area today. We quantify the evolution of geometry (length, area, elevation, slope), mass balance, and debris cover at a high spatial and temporal resolution since the end of the Little Ice Age (LIA) around 1850. We use these data to investigate the relation and interactions between geometry evolution, ice flow, and debris cover. We further analyse the occurrence of ice cliffs – typical surface features of debris-covered glaciers – and their role for the long-term glacier evolution.

**2   Study Site**

Zmuttgletscher is located in the southern end of the Mattertal in the western Swiss Alps and ranges from ~2240 m to ~4150 m a.s.l. It is surrounded by the Matterhorn (45.976N, 7.659E; 4478 m) and the Dent d'Hérens (45.970N, 7.605E; 4174 m) to the south and the Dent Blanche (46.034N, 7.612E; 4357 m) to the north, and high ridges (commonly above 3500 m) in between (Figure 1). At present, the glacier has a surface area of ~16 km² with a substantial part of debris cover in its ablation area,

originating from the surrounding rock walls.



Zmuttgletscher lies in the area of the Dent-Blanche nappe, which is part of the East-Alpine nappe. Most summits and rock walls consist of slightly metamorphosed crystalline and magmatic rocks (various types of gneiss, gabbros and diorites (Gouffon and Bucher, 2003).

Zmuttgletscher is located just north of the divide of the main Alpine ridge. The Rhone and Aosta valleys a few kilometres to the north and south, respectively, are well shielded by high mountains to the north and west and belong therefore to the driest regions in the Alps with precipitation values around 600 mm (Grächen, Mattertal: 653 mm; MeteoSwiss, 2017). Precipitation is mostly linked to northern and southern weather situations with a strong western component. No direct measurements at higher elevations are available; model estimates suggest values between 800-1500 mm in the area of Zmuttgletscher (MeteoSwiss, 2014). Glaciological mass balance measurements at near-by almost debris-free Findelgletscher show end-of-winter accumulation around 0.8-1.5 m water equivalent (w.e.) (Sold et al., 2016) and similar values can be assumed for Zmuttgletscher. However, avalanching importantly contributes to accumulation. Including contributing rock walls and steep lateral moraines, the total area available for accumulation that is eventually being transported onto the ice is up to 22 km² (restricted to areas of >30° slope angle).

Zmuttgletscher is a system of several independent and connected tributaries (Figure 1): The major accumulation area in the south – Tiefenmattengletscher (TMG) – reaches almost up to the summit of Dent d'Hérens; the western accumulation area – Stockjigletscher (STG) – defines a relatively flat area above two distinct ice falls between Tête Blanche (3707 m), Tête de Valpelline (3799 m) and Stockji (3092 m); Schönbielgletscher (SBG) is a tributary from the north reaching up to 3400 m, below the Dent Blanche summit wall. The upper part of TMG is completely surrounded by rock walls, including the >1000 m high Dent d'Hérens north wall and the almost 1500 m high Matterhorn west face. At the end of the LIA, the main glacier tongue was nourished by contributions from all of these accumulation areas. Today, it is mainly fed by ice from TMG and parts of the STG accumulation area. The ice flow from STG accumulation area is split into a part flowing to the north-east over a steep ice fall at ~2900 m, and a part flowing south-east, which is again split by a nunatak (point 2905); since ~2010 the small branch's lower end north of this nunatak is detached from the main glacier tongue. Because STG has a relatively flat accumulation area not framed by high walls, it does not deliver substantial amounts of debris down to the ablation area. SBG is located relatively low but receives abundant additional accumulation through avalanching from surrounding rock walls that are between 200 m and 900 m high. Consequently, its extensive debris cover exhibits a continuous surface layer even above the ice fall at ~2900 m.





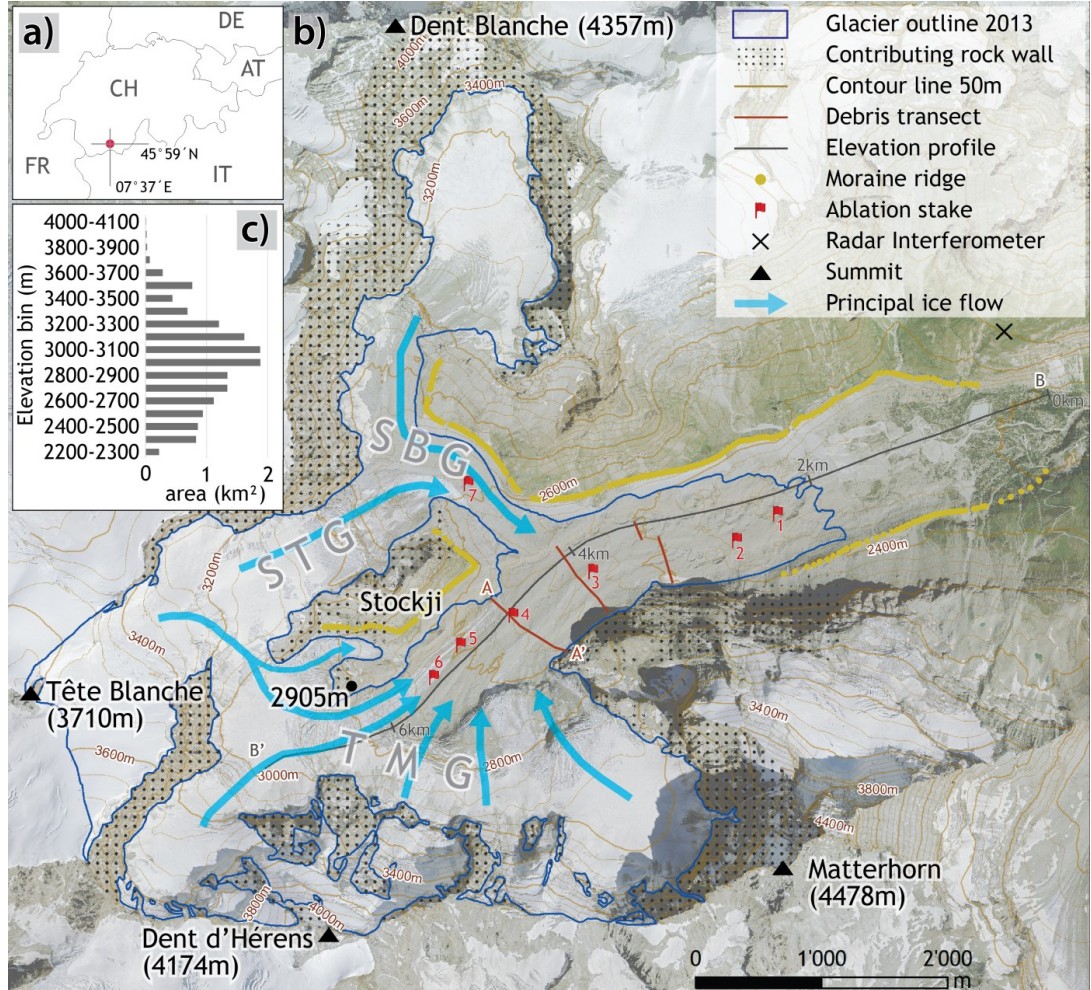

**Figure 1: a): Geographical location. b): Zmuttgletscher and its topographical setting. SBG = Schönbielgletscher, STG = Stockjigletscher, TMG = Tiefenmattengletscher. Background image: satellite orthomosaic by Swisstopo from 2013 (Swisstopo, 2010). c): The glacier hypsography (2010).** The glacier was part of the regular length monitoring programme of the Swiss glacier

monitoring network (GLAMOS) until 1997, when it was abandoned due to the uncertain position of the debris-covered, stagnant terminus. Furthermore, dendrochronological analysis tried to reconstruct former historic advances (Schneebeli and Röthlisberger, 1976), and an early study by Haefeli (1952) investigated ice deformation.

## 3   Data and Methods

The analysis of this study was based on input from several sources: topographical maps, plane-based and UAV-based aerial

images (Table 1), satellite images, various field observations, and long-term temperature measurements. The use of the data is discussed in the respective sections below.



**Table 1 : Topographic maps, satellite and aerial images used in the study. Abbreviations: OP = orthophoto, obl. aer. = oblique aerial, Plé. = Pléiades. Entries denoted with * are used as a final product. All data except 1894, 2016 and 2017 from Swisstopo (2018b). Photos from 1894 taken by Reid (1894).**

| Date | Product | Source | Used for dH tongue | Geod. MB | Debris cover | Velocity | Vertical uncertainty (m) | Spatial resolution (m), DTM / OP |
|---|---|---|---|---|---|---|---|---|
| 1859 | Map | Dufour map | | | x | | | Hand-drawn |
| LIA (~1850) | DTM | Moraine trimlines | x | | | | | 12 |
| 1880 | DTM, map | Siegfried map | x | x | x | | -7 | Hand-drawn (12m) |
| 1894 | Photo* | 2 obl. aer. photos* | | | x | | | |
| 1930 | Photo* | 1 obl. aer. photo* | Used for surface features interpretation. | | | | | - |
| 1946 | DTM, OP | Aer. Stereo | x | x | x | x | 5 | 8 / 0.5 |
| 1961 | DTM, OP | Aer. Stereo | x | | x | x | 1 | 2 / 0.5 |
| 1977 | DTM, OP | Aer. Stereo | x | x | x | x | 1 | 4 / 0.4 |
| 1983 | DTM, OP | Aer. Stereo | x | | x | x | 1.42 | 2 / 0.15 |
| 1988 | DTM, OP | Aer. Stereo | x | x | x | x | 1.19 | 5 / 0.35 |
| 1995 | DTM, OP | Aer. Stereo | x | | x | x | 0.77 | 1 / 0.15 |
| 2001 | DTM, OP | Aer. Stereo | x | x | x | x | 1.39 | 1 / 0.35 |
| 2005 | DTM, OP | Aer. Stereo | x | | x | x | 0.88 | 2 / 0.5 |
| 2005 | OP* | Aer. Stereo* | | | | x | | - / 0.25 |
| 2007 | OP* | Aer. Stereo* | | | | x | | - / 0.25 |
| 2010 | DTM, OP* | Aer. Stereo* | x | x | x | x | 2 | 2 / 0.25 |
| 2013 | OP* | Aer. Stereo* | | | | x | | - / 0.25 |
| 2016 | DTM, OP | UAV stereo | x | | | x | 1.24 | 0.5 / 0.25 |
| 2017 UAV | DTM, OP | UAV stereo | x | | | x | 1.13 | 0.5 / 0.25 |
| 2017 Plé. | DTM | Sat. Stereo | x | x | | | 0.81 | 1 / - |

## 3.1 DTM and orthophoto generation

Glacier surface information was extracted from stereo-photogrammetric DTMs generated from aerial images, a well established method to reconstruct elevation changes (e.g. Agata & Zanutta 2007). Aerial images were available from different sources (Table 1). They include

- Post-war mapping flights by the American military (Swisstopo 2018)
- National flight campaigns for topographic map production
- Specific flight campaigns for glacier monitoring purposes (Swisstopo 2018) conducted by Swisstopo
- Fixed-wing UAV flights (using a SenseFly eBee) 09/2016 and 10/2017
- Pléiades tri-stereo images from 09/2017

The time series of DTMs from 1946 to 2017 (apart from 2010 and the Pléiades DTM 2017) was created with photogrammetric methods using Structure-from-Motion software packages (Agisoft LLC, 2016; Pix4D, 2016; cf. Mölg and Bolch, 2017). The quality of the DTMs is comparable to DTMs from traditional photogrammetric software. Their uncertainty – defined by the standard deviation over stable terrain around the glacier tongue – mostly lies within 2 m depending on the resolution of the aerial images, their number and image quality factors (somewhat higher values were obtained for the DTMs from 1879 and

1946; Table 1). The uncertainties were derived for terrain with comparable steepness to the glacier surface (<30°) and are assumed constant in space, although DTM quality decreases in steep areas (e.g. rock walls surrounding the glacier). The DTM from 2010 (SwissAlti3D) was taken as a final product from Swisstopo and was also produced from aerial stereo images; it has a nominal resolution of 2 m and a vertical accuracy of 2 m (Swisstopo 2018). In addition, we generated a glacier-wide DTM for 2017 using high-resolution satellite imagery from Centre national d'études spatiales – Pléiades, with a ground resolution

of 1 m. The tri-stereo image was produced based on photogrammetric principles using PCI Geomatica (Geomatica 2016). All



DTMs were co-registered before further analysis by using the analytical approach by Nuth and Kääb (2011), followed by a second order trend surface correction to eliminate remaining elevation differences (Pieczonka et al., 2013).

Orthophotos were generated by rectification of the stereo images using the respective DTM. These images were subsequently used for delineating the glacier boundary and mapping of exposed ice. We further used the Swissimage by Swisstopo –
orthorectified image data from aerial imagery – that was available for the years 2005, 2007, 2010, and 2013 (Swisstopo 2018).

### 3.2   Glacier area and length

Glacier area was extracted from the Dufour map (1859), the Siegfried map (1879), and all available orthophotos (see Table 1) by manual digitizing. The Dufour map (map sheet 22, section 8, number 485) dating back to 1859 is the first map containing elevation information (in the form of contour lines) and distances acquired with modern methods (Wolf, 1879; Graf, 1896;
Rastner et al., 2016). The extent of the glacier and the supraglacial debris could be extracted from the map, whereas its elevation information was disregarded due to strong, non-linear, horizontal distortions. The Siegfried map in 1880 was a follow-up, containing elevation information at Zmuttgletscher from 1879 (Swisstopo, 2018a). The map was georeferenced using ground control points (GCPs), i.e. reference points indicated in the map that could be referred to stable points today.

The hanging glaciers at the north face of Dent d'Hérens have been included in the glacier area, since they contribute mass to
the main glacier through roughly regular 'ice fall' events. This resulted in a glacier area information for each date since 1859. The mapping quality is commonly lower in debris-covered compared to debris-free areas (Paul et al., 2013), but the high resolution of the images allows for correctly interpreting the glacier margin. The glacier boundary in the accumulation area of STG to Glacier de Ferpècle and Haut Glacier de Tsa de Tsan was taken from the 2010 ice divide and was kept stable over time.

Front variation measurements were conducted by using the glacier outlines for each date. Along the ice front – perpendicular to the flow – the change was measured at distances of 100 m and then averaged (Koblet et al., 2010; Bhambri et al., 2012). For the comparison to other glaciers we used GLAMOS length variations (GLAMOS, 2018), which were acquired in the field with the same concept of several parallel measurements, equidistant by 50 m. At Zmuttgletscher, GLAMOS measurements have been conducted almost annually from 1892 until 1997, an additional data point was added for 2010. To properly interpret
the length change information, we compared the variations from our measurements to the ones by GLAMOS, and finally compared length variations over time from various Swiss glaciers.

The uncertainty of both area and length results are estimated to lie within $\pm^{1}/_{2}$ pixel along the glacier boundary or at the start and end of the length profile (Hall et al., 2003; Granshaw and G. Fountain, 2006; Bolch et al., 2010; Bajracharya et al., 2015). For the calculation of the changes, the uncertainties of the two respective dates are combined by error propagation, analogue
to the mass balance data (Hall et al., 2003).

### 3.3   Debris cover, on-site ablation and on-site temperature

Debris cover extent was manually digitized using the orthophotos as well as the two historic maps which already contain a debris cover symbol (Figure 2a and b). Further, the debris extent of the Siegfried map (1879) could be verified using two photographs taken in 1894 (Figure 2c). This information is valuable to limit the debris extent at and up-glacier of the confluence
of TMG and SBG, which is the region of the strongest changes.





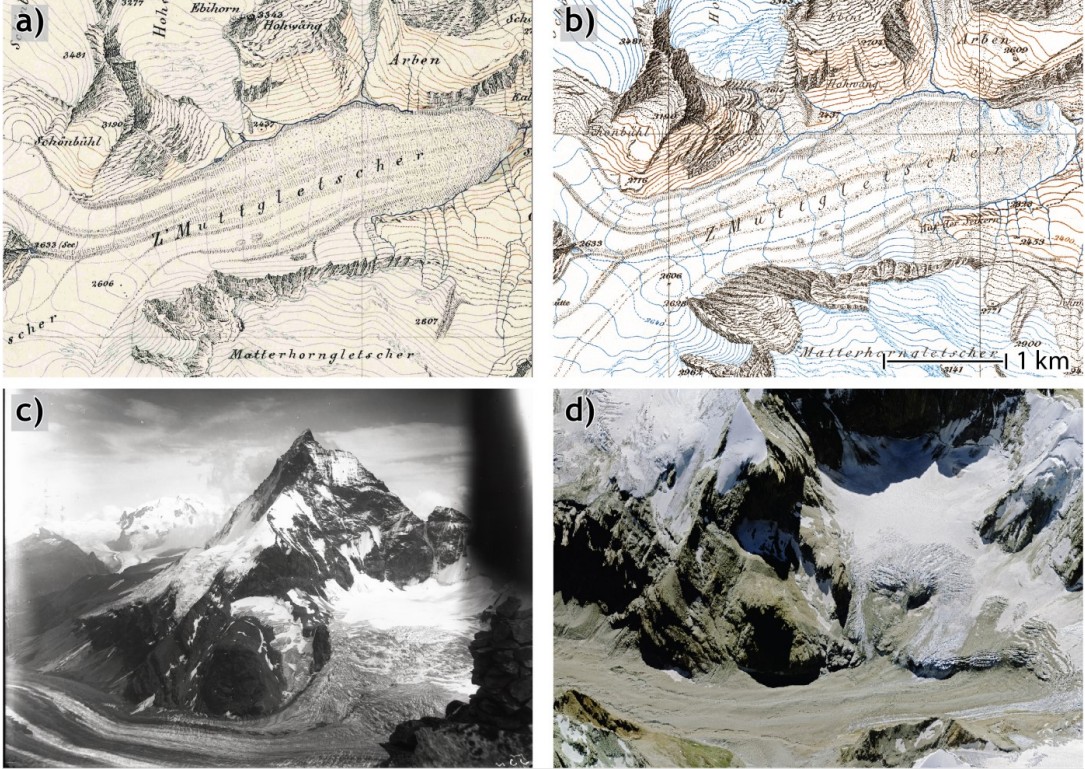

**Figure 2: Example of mapped debris cover on (a) Dufour map, 1859, and (b) Siegfried map, 1879. Panel (c) shows parts of the tongue in 1894 in direction South-West (Reid, 1894; National Snow and Ice Data Center, 2002, updated 2015). d: similar extent as in c) in 2013.**

We undertook ablation measurements at several points on the glacier tongue during summer 2017 for setting the elevation change observations into context. These point measurements over two years were conducted on bare ice and on surfaces with variable debris thickness using PVC stakes of 2 m length that were connected by zip ties and drilled into the ice. Debris cover was removed for the drilling and repositioned after inserting the stakes. To understand the effect of the ice cliffs on ablation, the horizontal backwasting of a south-facing and north-facing ice cliff was measured using horizontally placed ablation stakes.

The stakes were measured in intervals of several weeks over the course of the summer, typically from end of June to mid October.

Debris thickness information was collected in the field along several transects across the glacier tongue with additional single measurements in between by manual digging (for transect locations see Figure 1; results of transect see Figure 7). Each data point represents the average of three measurements ~1 m apart.

Homogenised time series of temperature measurements by MeteoSwiss (Füllemann et al., 2011) were used to put the observations into context. Stations had to be close, cover a long period, and be situated at the elevation of the glacier tongue. Since no single station fulfilled all requirements, we used three stations to cover these criteria: Sion, Zermatt, and Col du Grand St. Bernard.

### 3.4    Surface features

Ice cliffs, water flow channels from supraglacial melt streams with exposed ice, and supraglacial lakes were extracted in a semi-automatic process using an object-based approach with Trimble eCognition (eCognition Essentials 1.3, 2016). The respective location and area was determined on all orthophotos from 1946 to 2013. A primary segmentation divides the image into polygons based on pixel intensity and image texture. Ice cliffs often consist of a lower, steeper section of bare ice, and a



flatter section of ice covered with sand and pebble-sized debris particles in the upper part, where the maximum solar angle seems to define the slope of pole-facing cliffs (Sakai et al., 2002; Buri et al., 2016). These changing slope areas were often separated by segmentation, and were in a second step manually selected and combined into one polygon per ice cliff or lake. Using the approach above, water flow channels and ice cliffs could not be mapped as separate features and are summarised

under the category 'exposed ice area'. The described approach is effort efficient and assures a low uncertainty level, which is in the order of ±½ pixel.

In order to assess the importance of exposed-ice areas with respect to surface elevation change, we investigated the changes at the lower section of the glacier tongue over time. A 35 m buffer was generated around the exposed-ice polygon for both date 1 and date 2, and the overlapping area of these two buffer zones defined the category of exposed-ice areas. For each period the

average surface elevation change in this category was compared to the surface elevation change at the rest of the tongue (yellow area in Figure 8).

### 3.5    Reconstructing the glacier surface topography for the 19th century

At the end of the LIA many Alpine glaciers reached a maximum Holocene extent, similar to the LIA maximum around 1650, and accordingly moraines were built up or former high stands were again reached, approximately in the 1850s (Steiner et al.,

2008; Zumbühl et al., 2008). Moraine elevation is available for large parts around today's ablation area of Zmuttgletscher (indicated in Figure 1). The maximum elevation of lateral moraines was used to interpolate the glacier surface elevation using the Natural Neighbour method (Sibson, 1981); we assumed a plane across the glacier surface due to the small curvature indicated in the Dufour map (Graf, 1896; Swisstopo, 2018b). This LIA surface elevation was not used for calculating geodetic mass balance because of the incomplete areal coverage.

The contour lines of the Siegfried map were semi-automatically digitized by separating the differently coloured (blue = on-glacier, brown = off-glacier) contour lines from other symbols (Siedler, 2011). Their elevation information was extracted and interpolated using ArcGIS´ Topo-to-Raster tool to achieve a distributed DTM across the glacier tongue. The point of origin for elevation measurements, situated at the shore of Lake Geneva, changed in the 1930s from 376.2 m to 373.6 m (Swisstopo, 2018b) and the elevations derived from the Siegfried map were corrected accordingly. Elevation information from the Dufour

map was disregarded due to strong, non-linear, horizontal distortions.

### 3.6    Surface elevation changes and geodetic mass balance

The time series of surface elevation change over the glacier tongue was restricted to the overlapping area of all available DTMs (except the moraine-derived LIA DTM), from 1879 to 2017 (11 dates), and reaches up to ~2750 m a.s.l. and covers ~20% of today´s total glacier area. The surface elevation change maps were generated and investigated for each individual period as

well as for the entire time span. The LIA DTM yielded a mean difference of +7.3 m to 1879 with deviations of ±30 m compared to the Siegfried map. The strong similarity to the Siegfried map made us confident that the surface elevation values from 1879 are reliable. Because of the unclear timing (maximum stand of the LIA?) the LIA DTM was not used for surface elevation change analysis.

Geodetic mass balance was calculated for periods between dates where the DTMs covered most parts of the glacier surface

(1879, 1946, 1977, 1988, 2001, 2010, 2017). Due to DTM insufficiencies (artefacts, voids), certain areas were not covered, especially in higher elevations, and no surface elevation change values could be calculated there. These missing values were filled using a range of methods: (1) a linear relationship between elevation and elevation change (e.g. Kohler et al., 2007; Kääb, 2008; which is based on the strong respective correlation up to $R^2$=0.93); (2) using the same relationship but additionally setting the elevation change values to 0 when they become positive in the highest elevations; (3) using the mean elevation

change value of the uppermost elevations that contain data and applying it to the glacierised area above. Independent of the method it was necessary to work with elevation bins of 100 m (starting from 2100 m) to get a representative value on which





to base the calculations. Reason for this was the incomplete coverage of some glacier parts and thus a certain susceptibility to outliers, which could be minimised by the use of elevation bins. The average elevation change per elevation bin was multiplied with the area share of this bin of the total glacier area and summed up to reach the average elevation change of the total glacier surface. We assumed an average density of 850 ±60 kg/m³ (Huss, 2013) to convert volume to mass changes.

### 3.6.1 Uncertainties

The total uncertainty of the surface elevation estimates and the mass balance consists of (i) the accuracy of the individual DTMs, (ii) the filling of the empty elevation zones, (iii) the glacier volume density changes and density conversion, (iv) the debris volume changes, and (v) the DTM co-registration. Ice density changes are on average assumed to be negligible over a longer time span (Huss, 2013). As no information is available about changes in snow and firn volume, we assumed zero change for that, but included an uncertainty of ±60 kg/m³ for the density to mass conversion analogue to Huss (2013). Because not all DTMs used for mass balance calculation covered the entire glacier surface, the missing surface area had to be filled with values from another point in time, in this case the reference DTM from 2010. This procedure introduces an uncertainty since the area of the elevation bin might have changed over time, but the order of change is small due to the absence of big topographic steps in the glacier hypsography (Figure 1c). The error from the filling of the elevation zones is taken into account as the standard deviation between the total mass balance after applying the three different interpolation methods ($\sigma_{fill}$). This uncertainty measure was combined with the uncertainties of the two DTMs framing the respective period ($\sigma_{DTM1}$ and $\sigma_{DTM2}$, standard deviation over stable terrain) to be used as the total uncertainty for the resulting mass balance by applying the law of error propagation (e.g. Zemp et al., 2013) and hence given by

$$\sigma_{total} = \sqrt{\sigma_{DTM1}^2 + \sigma_{DTM2}^2 + \sigma_{fill}^2} \tag{1}$$

Uncertainties range from ±0.12 to ±0.2 m w.e./yr, whereas the uncertainty for the total period 1879-2017 is ±0.03 m/yr.

### 3.6.2 Surface elevation change for different debris thickness classes

On the time scale of several years, the effect of various debris thicknesses may impact on surface elevation changes. Using the debris thickness observations from field measurements, we defined debris thickness classes of thin debris (~5 cm) and thick debris (>15 cm) as well as of bare ice. We mapped representative areas of these classes for several periods between 1995 and 2017 (1995-2001, 2001-2005, 2005-2010, 2016-2017) and within these areas of ~20.000 m² we calculated the average elevation change values for each period.

### 3.7 Surface flow velocities

Surface flow velocities provide important information about the glaciers' dynamical state and its change over time. Automatic feature tracking methods were not feasible for the time before 2005 because the time differences and, thus, displacements of the features were too big. Therefore we manually tracked boulders to infer flow velocities along the debris-covered part of the tongue of Zmuttgletscher as well as on the lower branch of SBG. The tongue was divided into eleven sections according to differences in dynamic state and ice flow units. The individual measurements were averaged for every time period and section, respectively, to achieve a comprehensive picture of the dynamic changes (Figure 12). For the periods 2005-2007 and 2016-2017 we extracted flow velocity fields using the IMCORR module in SAGA GIS (Fahnestock et al., 1992; Scambos et al., 1992; Conrad et al., 2015). The results were filtered using a visually defined threshold of correlation quality ('Strength'). Outliers had to be manually removed, e.g. in the area of strong ice cliff change or pro-glacial water surfaces.

On 22nd-24th of August 2017 the tongue of Zmuttgletscher was observed additionally with a terrestrial radar interferometer (GPRI) developed by GAMMA Remote Sensing and Consulting AG, Switzerland. The GPRI was installed on a hill about 3 km away from today´s terminus (Figure 1). Measurements were acquired every minute for 1.5 days with a final range



resolution of 3.75 m and an azimuth resolution of 7 m at a slant range of 1 km. The interferograms were determined with a standard workflow following Caduff et al. (2015) using the Gamma software. The interferograms are stacked over a window of 8 hours to reduce noise and afterwards unwrapped using a stationary point on the main land. The unwrapped phases were then converted to line-of-sight displacement according to Werner et al. (2008), whereby the negative velocities are considered
as noise and filtered out. The results were georeferenced by rotating through an estimated angle based on the best match with the DEM25 (Swisstopo, 2005). Afterwards, the data was resampled into the new grid using nearest neighbour interpolation. To access the uncertainties in the velocities we looked at the difference from zero in measured displacement of 10 stationary points. This results in an uncertainty of the stacked velocity maps of ≤ 0.03 m/day.

## 4    Results

### 4.1    Glacier area and length changes

Zmuttgletscher retreated 1907 ±12 m (i.e. 12.1 ±0.09 m/yr) since close to the end of the LIA (~1850s) until 2017 (Figure 3). With 21.7 ±0.04 m/yr, the retreat rate was strongest between 1961 and 1977. After 1977 until 2001 the glacier terminus stagnated or even slightly advanced. Between 2001 and 2005 Zmuttgletscher accelerated its retreat reaching 19 ±0.15 m/yr, and slightly slowed down since 2005 to 12.1 ±0.06 m/yr. The total glacier area decreased from 21.2 ±0.34 km² in 1859 to 15.7
±0.07 km² in 2017 (Figure 4). Similar to the length change, the area decreased only slightly between 1977 and 2001 (-0.01 ±0.006 km²/yr), after which the loss rate increased again to -0.06 ±0.004 km²/yr.

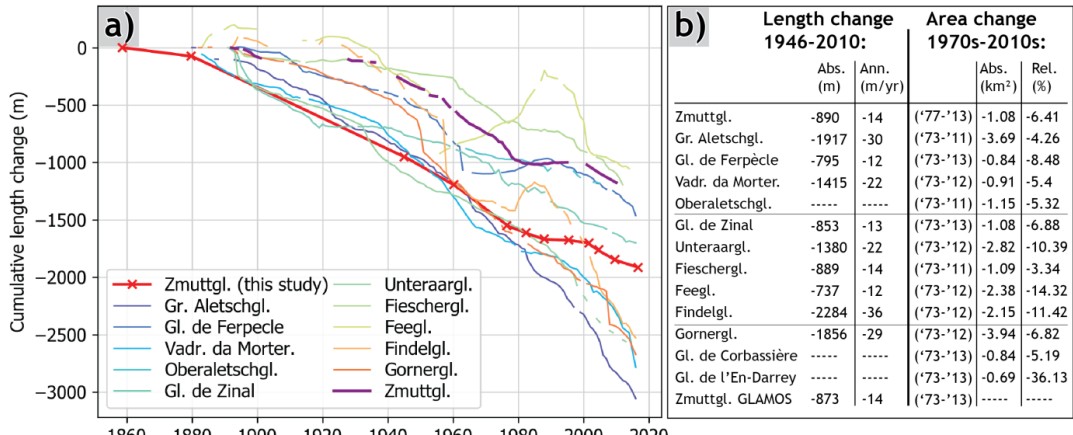

| | Length change 1946-2010: | | Area change 1970s-2010s: | | |
|---|---|---|---|---|---|
| | Abs. (m) | Ann. (m/yr) | | Abs. (km²) | Rel. (%) |
| Zmuttgl. | -890 | -14 | ('77-'13) | -1.08 | -6.41 |
| Gr. Aletschgl. | -1917 | -30 | ('73-'11) | -3.69 | -4.26 |
| Gl. de Ferpècle | -795 | -12 | ('73-'13) | -0.84 | -8.48 |
| Vadr. da Morter. | -1415 | -22 | ('73-'12) | -0.91 | -5.4 |
| Oberaletschgl. | ----- | ----- | ('73-'11) | -1.15 | -5.32 |
| Gl. de Zinal | -853 | -13 | ('73-'13) | -1.08 | -6.88 |
| Unteraargl. | -1380 | -22 | ('73-'12) | -2.82 | -10.39 |
| Fieschergl. | -889 | -14 | ('73-'11) | -1.09 | -3.34 |
| Feegl. | -737 | -12 | ('73-'12) | -2.38 | -14.32 |
| Findelgl. | -2284 | -36 | ('73-'12) | -2.15 | -11.42 |
| Gornergl. | -1856 | -29 | ('73-'12) | -3.94 | -6.82 |
| Gl. de Corbassière | ----- | ----- | ('73-'13) | -0.84 | -5.19 |
| Gl. de l'En-Darrey | ----- | ----- | ('73-'13) | -0.69 | -36.13 |
| Zmuttgl. GLAMOS | -873 | -14 | ('73-'13) | ----- | ----- |

**Figure 3: a): Cumulative length change of Zmuttgletscher with data points (red crosses) from this stud, and a number of other Swiss glaciers (GLAMOS, 2018). Zmuttgletscher shows a relatively modest retreat while other debris-covered glaciers – Unteraargletscher**
**and Glacier de Zinal – have retreated more rapidly. b): Length and area changes for a selection of Swiss glaciers. Area changes have been mapped on Swissimage and compared to the Swiss Glacier Inventory 1973 (Müller et al., 1976; Paul, 2004). For a complete list of annual retreat rates see Figure S11.**

### 4.2    Debris cover

#### 4.2.1    Debris cover evolution

Zmuttgletscher was with 2.8 ±0.2 km² modestly debris-covered at the end of the LIA (~12.9% of the entire glacier area). Until 2013 this area has increased to >5 km², while the total glacier area has decreased from >21 km² to <16 km², resulting in the debris covering ~33% of the total glacier area today (Figure 4).





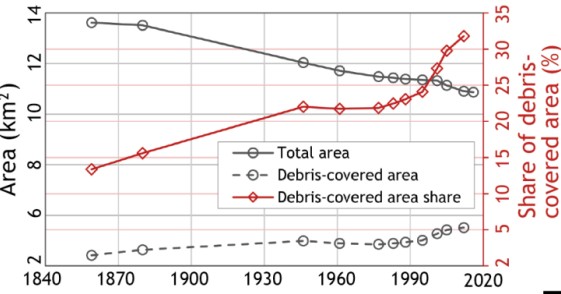

**Figure 4: Evolution of total glacier area, debris-covered glacier area, and the share of the debris-covered glacier area of the total glacier area.**

Generally, the debris cover extent has expanded up-glacier in all parts of the glacier. The extension was very pronounced along
the surface of SBG and in the central part of the main glacier tongue (Figure 5). In both areas the debris has expanded to above
the ice fall into the former accumulation areas and starts now close to the contributing rock walls of Dent d'Hérens. The debris
cover grew strongly also at the glacier margins below Stockji, due to further input from moraines and the disconnection of
contributing tributaries. Below the ice fall of STG, a small rock fall occurred between 2010 and 2013 (indicated in Figure 5)
after the exposure of the rock wall, covering an area of approx. 170x300 m. This debris mound is now slowly transported
down-glacier, while spreading laterally due to preferential ablation and fluvial transport.

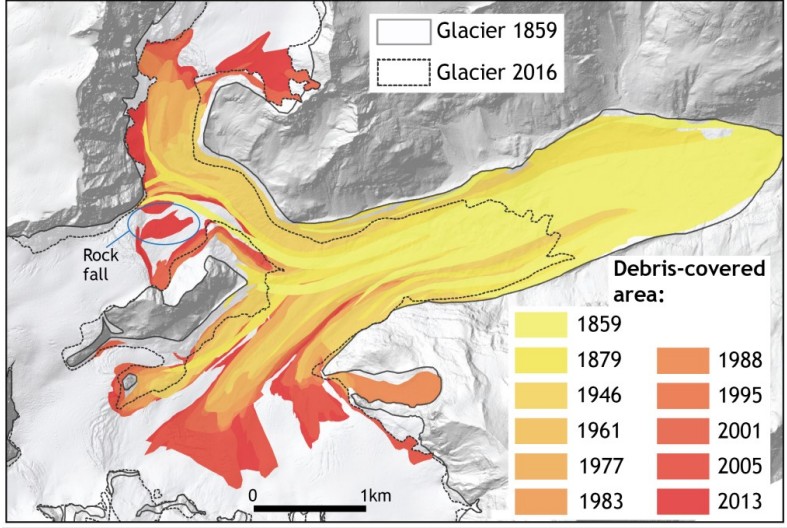

**Figure 5: Current extent of debris cover and its evolution since ~the end of the LIA (1850s) together with the glacier outlines for LIA and 2016. Even though the glacier has retreated in the debris-covered region, debris covers more and more of the total glacier area.**

**4.2.2    Debris cover characteristics**

Field investigations reveal a homogeneous debris cover in some regions with superficial stone sizes mainly between 10 and
cm diameter, and much more heterogeneous in other regions, with pebble-sized stones to boulders in the metre scale (Figure
6). In most regions the debris consists of more than one 'layer' (i.e. stones lie on top of each other). In such cases the debris is
sorted: fine-grained material (sand and even silt) lies directly on the ice, followed by a few centimetres of pebbles; above these
20   typical base layers follow larger stones without any more specific sorting. The rock walls' different geology leads to differently
coloured debris material, which has been deposited on the same location and led to elongated, coloured debris bands along the
glacier tongue. The thickness of the debris layer along the transect varies from below 5 cm to over 70 cm. The thickest areas



were found on the elongated ridge, especially on the steep, southern slope (Figure 7a). The average thickness along the transect was 16.3 cm ±1.3 cm.

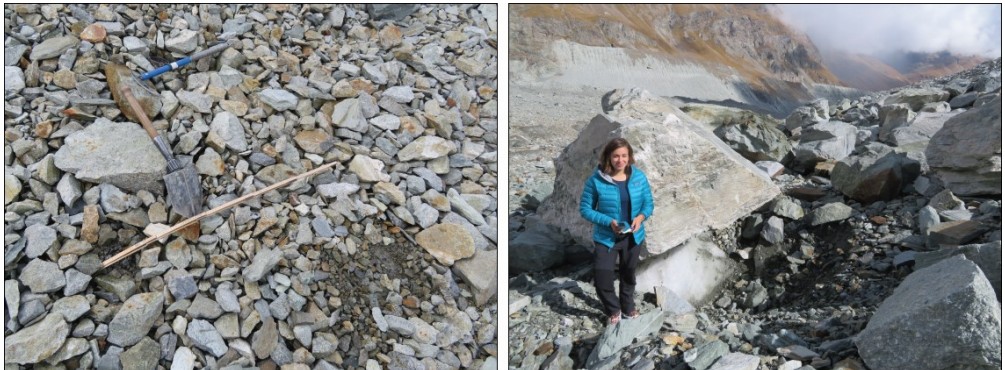

**Figure 6: Debris size and distribution on Zmuttgletscher: Homogeneous, small stones (left) vs. rocks and boulders (right). Scale in left is one metre.**

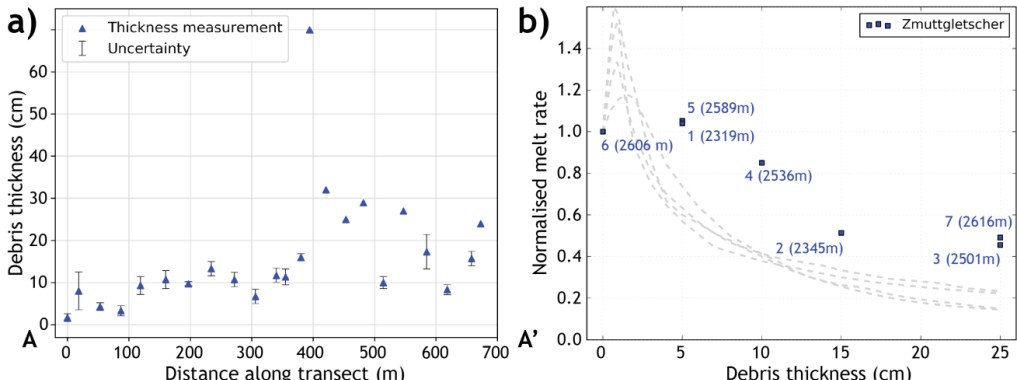

**Figure 7: a): Debris thickness along the transect indicated in Figure 1, ranging from A (=0 m) to A' (=700 m). Uncertainty bars represent the standard deviation of three measurements per location for depths <20 cm. b): Ablation measurements normalised by ablation on clean ice against debris thickness of seven stakes over the time period 05.07.-22.08. 2017. Stake locations are indicated in Figure 1. For comparison in dashed grey the curves of Rakhiot Glacier, Barpu Glacier, Kaskawalsh Glacier, and Isfallsflaciaeren (figure adapted from Mattson et al., 1993).**

Ablation measurements from seven locations and over several periods from August 2016 to October 2018 show an 'Östrem-like' behaviour with respect to debris thickness (Figure 7b). These results, from stakes at elevations between 2300 m and 2600 m indicate that melt is strongly dependent on debris thickness and less on elevation. Compared to the reference stake at 2606 m on debris-free ice, field measurements in summer 2017 (7 weeks) show a reduction of ~15% and ~50% for a debris thickness of 10 and 25 cm, respectively. Our results yield a somewhat smaller melt reduction by debris compared to literature values, where melt reductions of ~40-60% and 70-80% for ~10 cm and ~25 cm, respectively, were found (Östrem, 1959; Mattson et al., 1993; Nicholson and Benn, 2006; Brock et al., 2007).

### 4.3 Surface features and related elevation change

For the location and area of exposed ice we did not find any clear trend in number, area or location (Table 2). Areas of exposed ice are mostly located in the lower parts of the glacier tongue until the terminus (yellow area in Figure 8). The area covered by exposed ice decreased by over 40% in the 1970s and 1980s but increased again afterwards. The average slope of the exposed ice areas extracted from the DTMs is ~26° with no trend over time, but the variations are rather a result of variable DTM resolution and quality. In contrast, own field measurements show slopes of 40-50° for the ice cliffs. Due to this relatively low slope (in comparison to field measurements and literature, e.g. Reid and Brock, 2014) and the small total area, also the



topographically corrected exposed-ice area amounts to less than 5‰ of the total glacier area, and less than 1.8% of the debris-covered glacier area (Table 2). Only very few small ponds have been detected (<5 per time step) and due to their small number and area they have not been further analysed.

**Table 2: Evolution of exposed ice area from 1946-2013. ´dc.´ = debris-covered.**

| Year | Cliff area (km²) | Glacier area (km²) | Debris-covered area (km²) | Cliff share of dc. area (%) |
|------|------------------|--------------------|--------------------------|----------------------------|
| 1946 | 0.073 ±0.007 | 18.07 ±0.265 | 3.98 ±0.010 | 1.43 |
| 1961 | 0.075 ±0.007 | 17.43 ±0.065 | 3.79 ±0.010 | 1.79 |
| 1977 | 0.047 ±0.006 | 16.96 ±0.135 | 3.71 ±0.005 | 1.04 |
| 1983 | 0.040 ±0.003 | 16.87 ±0.065 | 3.79 ±0.000 | 0.64 |
| 1988 | 0.079 ±0.008 | 16.77 ±0.165 | 3.87 ±0.005 | 1.20 |
| 1995 | 0.045 ±0.001 | 16.72 ±0.065 | 4.03 ±0.000 | 0.96 |
| 2001 | 0.065 ±0.006 | 16.66 ±0.065 | 4.55 ±0.005 | 1.07 |
| 2005 | 0.084 ±0.005 | 16.28 ±0.065 | 4.85 ±0.005 | 1.54 |
| 2013 | 0.059 ±0.004 | 15.82 ±0.015 | 5.03 ±0.010 | 1.10 |

Over the total period, the average elevation change over the exposed-ice areas was 1.2 times higher than over the rest of the tongue (Figure 8). During the 1970s and 1980s the thinning rates were, however, almost the same and close to zero whereas since 1988 the factor of increased elevation change around exposed-ice areas was between 1.5 and 1.7 consistently higher than on average. When considering average thinning rates over the entire glacier tongue, the inclusion of exposed-ice areas enhances

the surface lowering only by less than 5 %, and thus the presence of ice cliffs does not seem to affect the mass loss substantially.

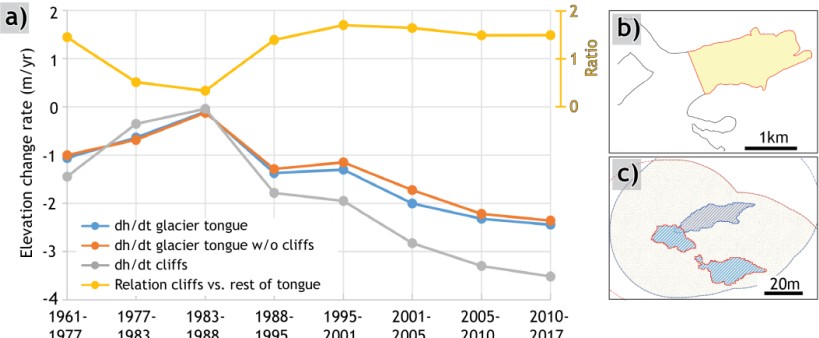

**Figure 8: Influence of ice cliffs on total elevation change rate over the lowest area of the glacier tongue for eight time periods from 1961 to 2017. a): elevation change rate periods; b): area that was considered as ´tongue´ for elevation change analysis; c): examples of ice cliffs of two different years (2005 = red; 2001 = blue), the corresponding borders of the 35 m buffer, and the overlapping area**
**of this buffer (yellow, grey dotted area).**

### 4.4    Surface elevation changes

#### 4.4.1    Tongue-wide surface elevation changes

Since the end of the LIA, the surface elevation of the Zmuttgletscher tongue below ~2750 m a.s.l. has almost continuously decreased (Figure 9). In some periods, mostly before 1977 (Figure 9a-d), the elevation change was strongest in the lowest

regions close to the terminus, in other periods, in particular since 2001, it was more heterogeneous throughout the tongue. The maximum values often exceeded -3 m/yr, whereas in the periods between the 1970s and 1990s some regions showed no or even positive elevation changes. Between 1879 and 1946, the surface elevation change was with -0.63 m/yr on average relatively small and most pronounced at the terminus. It intensified to -1.75 m/yr from 1946 to 1961. In the 1970s and 1980s a partial thickening of the glacier tongue is observed, which is most pronounced at the tip of STG tongue (+1.2 m/yr from

1977-1983) and in the main trunk of the tongue, while the terminus area was still substantially thinning. Between 1977 and 1983 the tongue's surface elevation was on average with -0.11 m/yr almost balanced. Between 1983 and 1988 the upper part of the tongue already started to thin again whereas downstream in the lowest 1 km a mixed signal of thickening and thinning



occurred between 1988 and 1995 (Figure 9f). After 1988 the tongue's surface elevation lowering reached again ~1-2 m/yr, increasing to >2 m/yr after 2005 and only at the tip of the tongue some positive values occurred between 1988 and 1995, consistent with a sligth advance at the centre of the terminus. On average, Zmuttgletscher tongue has thinned by 105 m since 1879, with maximum values of >190 m in the area of today's terminus position, i.e. the former central part of the tongue. High

5     local values of elevation decrease are observed in regions with high proportions of exposed ice or where no debris cover was present (panels f, h, I in Figure 9). Panels d and e of Figure 9 show a patchy pattern of surface elevation increase close to the terminus, which partly represent ice cliffs that were pushed down-glacier – at a higher rate than the ice cliff backwasting – by increased flow velocities.





**Figure 9: Elevation change over the glacier tongue: evolution since 1879. Areas of exposed ice are shown in red. Note how the 'wave' of mass flux moves down-glacier from c-f. High values in the forefield (e.g. 1983-1988) are linked to artefacts from water surfaces.**





### 4.4.2  Surface elevation change for different debris thickness classes

The average total elevation change for the period 1995-2017 was -13.6 m for bare ice, -6.8 m for thin debris, and -4.5 m for thick debris (uncertainty of ±1.1 m), which corresponds to thinning rates of 1.1 m/yr, 0.57 m/yr, and 0.38 m/yr, respectively. The relation of thick debris to bare ice ranged between 0.29 and 0.63 (average = 0.33). It is important to note that the different

debris thickness classes are – by nature – located in different elevations and, hence, do not share the exact same climate. With the UAV-derived DTMs from September 2016 and October 2017 the surface elevation change of one year in a resolution of 50 cm was calculated for the same glacier extent as displayed in Figure 9. The average elevation change in the overlapping area over the 13 months is -2.4 ±1.8 m but is very heterogeneous. It reveals along-flow structures in elevation change as well as spatial variation patterns related to ice cliffs and flow channels.

### 4.4.3  Evolution of the mass balance gradient

A glacier mass balance gradient is typically inclined towards lower elevations due to higher temperatures, whereas the insulation effect of a continuous debris layer can flatten this gradient. To estimate the strength of this debris cover effect, we divided the main tongue of Zmuttgletscher into several sections from 1 (terminus position of 1946) to 24 (highest glacier elevations of the overlapping DTMs, which are still to some extent debris-free today; Figure 10b). There is no clear hint of an

increasing balancing effect of debris cover over time (Figure 10a). A reversed gradient existed in the period 1983-1995, due to the 'wave' of additional mass flux travelling down-glacier, which is also the reason for the less negative and partly even positive values.

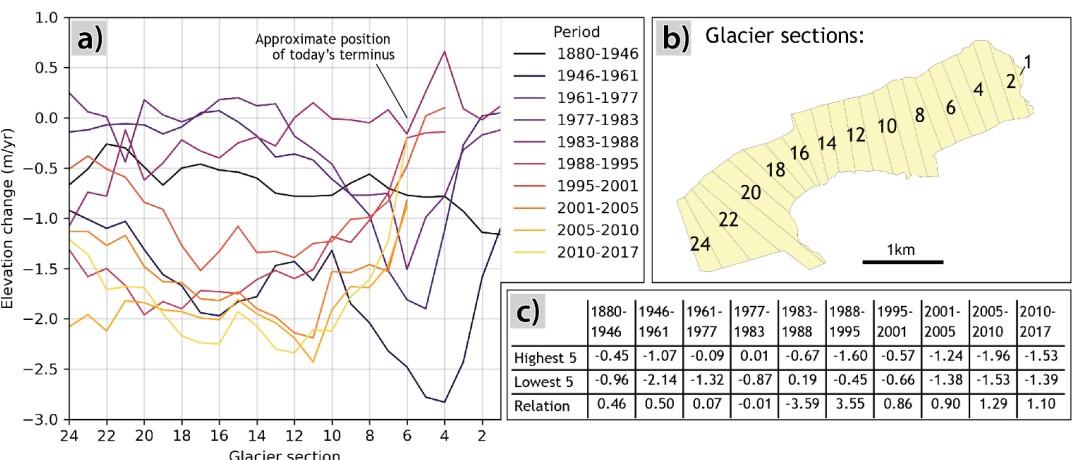

**Figure 10: a):** Average annual elevation change in different glacier sections. Early periods show a stronger elevation change towards
the terminus. In the 1980s this pattern changed and central sections (~8-20) have experienced the strongest thinning since then. In recent periods this pattern intensified. **b):** glacier sections from 1 at the terminus of 1946 to 24 at ~2700 m (today). **c):** Average elevation change values (in metres per year) of the five highest and lowest sections per period.

The average elevation change rate of the lowest and the highest five sections (disregarding the lowest sections after melt-out) yield the strongest contrasts; the relation between these two regions has changed over time and has been above 1 in recent
years (Figure 10c), even though the elevation change has reached high absolute values. We can discern a levelling of the elevation change in the lower part of the glacier tongue, which we attribute to the effect of debris cover, although the two lowest sections show still the strongest thinning.

### 4.5  Glacier mass balance

Zmuttgletscher's long-term mass balance from 1879 to 2017 has been negative (-0.31 ±0.03 m w.e./yr; Table 3). During the
climatically favourable period in the 1970s and 1980s Zmuttgletscher yielded a smaller mass loss, even below the Swiss-wide



average (Figure 11). However, although certain areas on the tongue showed an elevation increase (Figure 9c, d, e), the glacier-wide mass balance stayed negative (-0.27 ±0.18 m w.e./yr between 1977 and 1988). After 1988, the mass balance turned more negative again, even if the lowest areas on the tongue still showed stable or even increasing surface elevation (Figure 4f). The negative trend intensified and since 2001 Zmuttgletscher's mass balance has become more negative than the Swiss-wide average.

**Table 3: Surface elevation changes and geodetic mass balance of Zmuttgletscher.**

| Period | Annual mass balance (m w.e.) | Annual elevation changes (m) |
|---|---|---|
| 1879-1946 | -0.09 ±0.12 | -0.1 ±0.13 |
| 1946-1977 | -0.67 ±0.16 | -0.79 ±0.17 |
| 1977-1988 | -0.27 ±0.18 | -0.32 ±0.21 |
| 1988-2001 | -0.47 ±0.17 | -0.55 ±0.2 |
| 2001-2010 | -0.98 ±0.20 | -1.15 ±0.18 |
| 2010-2017 | -0.82 ±0.16 | -0.96 ±0.18 |
| 1879-2017 | -0.31 ±0.03 | -0.34 ±0.05 |

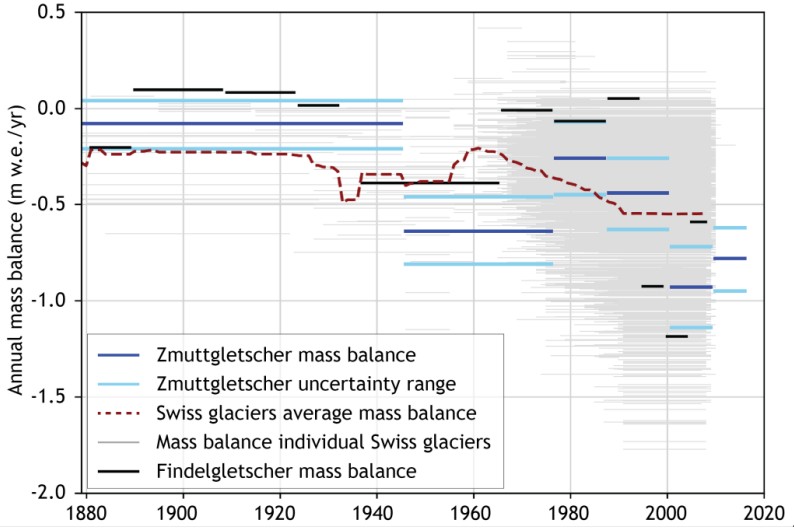

**Figure 11: Geodetic mass balance of Zmuttgletscher and Findelgletscher (data from Rastner et al., 2016) in comparison to all Swiss glaciers since 1879. Mass balance data from WGMS (2017).**

### 4.6    Surface flow velocities

Surface flow velocities were derived along the glacier tongue, from below the icefall of TMG, as well as along the debris-covered tongue of SBG. Generally and as expected, central parts of the glacier ('mid-left', 'mid-right') are faster than the margins, and velocities strongly decreased down-glacier. Overall, there is a trend of decreasing velocities since the first measurement period 1961-1977 (Figure 12) until today, but with a clear phase of acceleration in the late 1970s and 1980s. Velocities in the lower part of the glacier tongue (red in Figure 12) have decreased from 10-20 m/yr in the 1980s to almost zero (<3 m/yr since 2005). The central section (orange in Figure 12) showed values of 30-40 m/yr until the 1980s; afterwards velocities strongly decreased to ~5-10 m/yr today. The highest part of the tongue (blue) has decreased over the same period from 50-60 m/yr to ~15 m/yr. In the periods 1977-1983 and 1983-1988 a velocity increase of close to 50% was observed in all areas, with a slightly delayed signal further down on the lower tongue. It was followed first by a strong velocity decrease in the 1990s that got weaker thereafter. Since the 2000s the lowest part of the tongue (red) as well as the Lower SBG show values around 5 m/yr and smaller, which can be considered as almost stagnant.





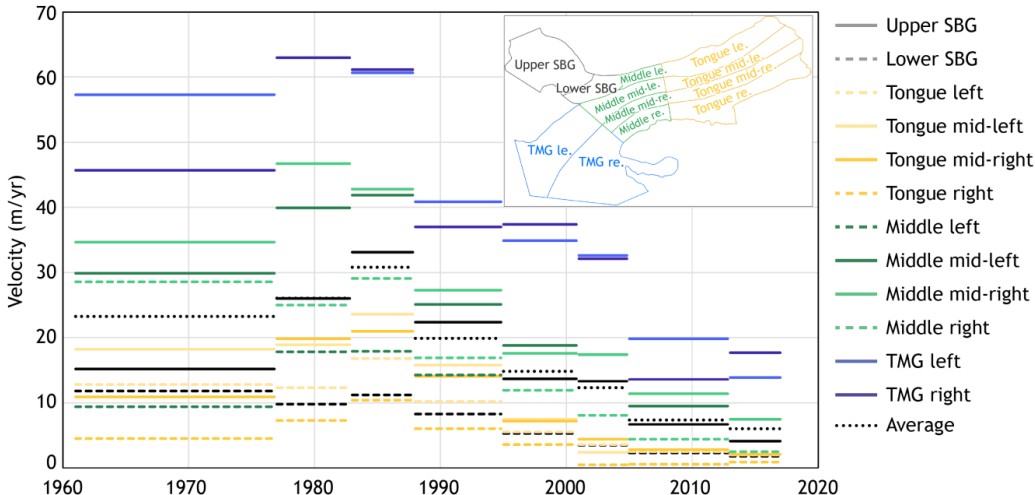

**Figure 12: Evolution of surface flow since 1961 for the different sections of the glacier tongue as labelled in the inset in the upper right corner. After an increase in the 1970s/1980s a rapid slow-down occurred in all observed regions.**

The radar interferometer displacements from the 1.5 day period in summer as an independent measurement yield similar results
and confirm the quasi-stagnation of the lower tongue with line-of-sight flow speeds <2 m/yr. The more active part of the glacier
with velocities >10 m/yr extends up-glacier from just above the confluence of TMG and SBG (orange and blue sections in
Figure 12). During recent years, the flow in this mid-tongue section has strongly decreased and velocities have halved between
2005-2007 and 2016-2017 (Figure 12 and Figure 13).

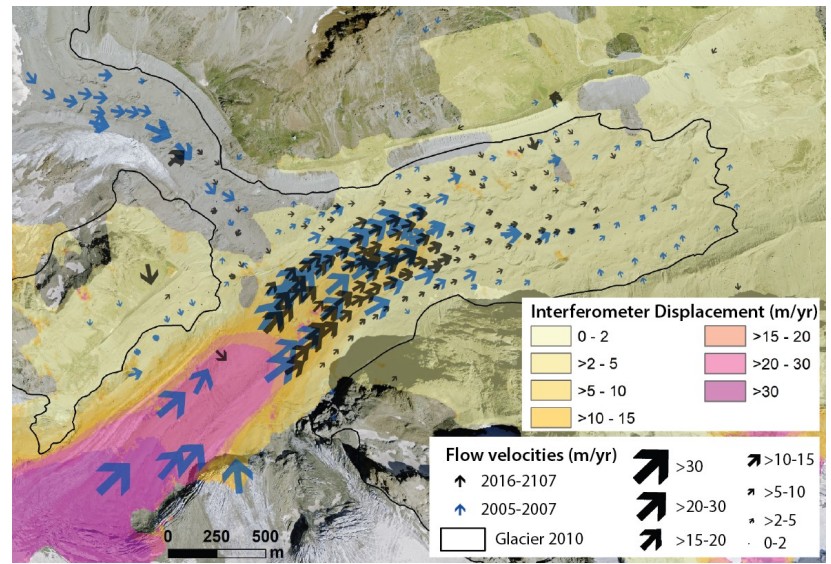

10  **Figure 13: Annual surface flow velocities from feature tracking for the periods 2005-2007 and 2016-2017 (arrows) and summer velocities from radar interferometer measurements on 22nd-24th August 2017 (yellow-pink coloured surfaces).**

## 5    Discussion and Interpretation

### 5.1    Methods suitability and shortcomings

The GLAMOS length change data (GLAMOS, 2018) coincides well with our observations for the period from 1946 to 2010
15  (difference of <3%, i.e. -863 m vs. -890 ±0.6 m). For the earlier period 1892-1946 we found a retreat of 727 ±12 m, which is





the cumulated annual retreat extracted from the linear interpolation between the positions of 1879 and 1946, whereas GLAMOS data shows a smaller retreat of only 303m. This deviation can be explained by (1) a potentially strong retreat between 1879 (our data point) and 1892 (first GLAMOS measurement), (2) a deviating terminus definition of the GLAMOS observer in 1892, or (3) an erroneous use of existing reference points during periods when measurements were not conducted

annually. In contrast to the GLAMOS field measurement time series, our results have been acquired with one consistent method, by the same observer and in the same study. All our data points originate from direct measurements on orthophotos, except 1879, which was extracted from the Siegfried map and contains the interpretation of the field analyst and the cartographer, introducing a – presumably small – potential error.

The DTMs used for calculating mass balance and surface elevation changes have uncertainties of ~1-2 m (i.e. ±0.2-0.4 m/yr),

which are derived as the standard deviation over stable terrain and rather represent the upper bound of the uncertainty (Magnússon et al., 2016). Except for the period 1977-1988 these uncertainties are significantly lower than the glacier changes and have thus little impact on the final decadal elevation change rates. High elevation change values outside of the glacier surface are found in very steep terrain and in the immediate forefield, which stem from artefacts on water surfaces (e.g. Figure 9 panel e), or the activities of a gravel extraction company (e.g. Figure 9g, h, i), and also do not influence the results on glacier

change.

The time series of glacier surface flow velocities along the lowest ~4 km is over 50 years long (1961-2017) and includes periods of glacier acceleration and deceleration. Even though it covers less than half of the total investigation period, it is one of the longest time series of a debris-covered glacier, similar to the series by Capt et al. (2016), who extracted velocities for debris-covered Glacier de Tsarmine from 1967 to 2005. Even longer time series using single boulder measurements were

performed and collected by Kirkbride (1995) and Haritashya et al. (2015) for insights into the evolution of Tasman Glacier since the 19[th] century. Although it was difficult to retrieve results for boulder tracking in more active areas on Zmuttgletscher due to large displacements (long periods between image dates) and crevasses (e.g. massive increase of crevasses between 1977 and 1988), several representative data points could be found per glacier sector and time period, and with this information we could draw a clear picture of the dynamic evolution of the entire glacier tongue since 1961. Unfortunately, the time period

1946-1961 yielded too few data points (due to image quality and snow cover) and was thus excluded from the time series. In the lowest areas the velocity field derived from the orthophotos 2016-2017 (Fig. S14) shows higher values than the interferometer (e.g. 4 ±0.25 m vs. 1 ±0.57 m/yr), indicating a smaller quasi-stagnant glacier part. This difference can be attributed to the larger uncertainties of the radar (which only records displacements in view direction) as well as to the season of measurements: especially in lowest areas flow velocities might be below annual average in August compared to late spring

or early summer, due to the formation of an effective englacial and subglacial drainage system.

Debris cover was manually mapped based on topographic maps and orthophotos. A continuous debris cover has reflection characteristics different from ice and can be distinguished visually. The transition from debris-free to debris-covered is continuous, and regions of sparse debris cover – which might even lead to increased ablation – were not included. We mapped only coherent areas of continuous debris cover, thus avoiding to include mixed pixels that indicate a sparse debris cover.

Because of varying properties of the orthophotos (RGB vs. panchromatic, resolution, contrast, texture) an automated method would not deliver equally satisfying results and was not applied. The uncertainty is equal to that of the glacier and ice cliff mapping, i.e. ±½ pixel. The debris cover of 1961 is less certain, because the orthophoto does not fully cover the ablation area and is additionally snow-covered in higher areas. Thus, the area might be somewhat underestimated, which would partly explain the decreasing debris-covered area between 1946 (3.91 ±0.01 km²) and 1961 (3.67 ±0.01 km²), nevertheless we think

that the observed trend is correct.

Mapping of exposed-ice areas is commonly done manually or by automatic detection including manual correction (e.g. Ragettli et al., 2016; Brun et al., 2016; Reid and Brock, 2014). The detection using only topographic and geometric variables showed no satisfying results, e.g. because many steep areas (>40°) are still debris-covered and many cliff areas show up less steep in




the DTMs. Our object-based mapping approach is efficient to map larger numbers of cliffs for several dates; however, the largest time share was still spent on manual selection of automatically delineated polygons, and there is a subjective choice of separation parameters and the selection of polygons. Depending on these aspects and the image quality (texture, contrast, snow cover, cast shadow) the results include both omission and commission errors, which also lie in the range of $\pm^1/_2$ pixel along

the polygon margins.

### 5.2    Explanation of glacier evolution

#### 5.2.1    Chronological evolution and interactions

The climatic history since the end of the LIA has impacted all glacier-related variables. In the 1940s and 1950s positive temperature anomalies and long droughts (Hirschi et al., 2013) led to negative mass balance for most glaciers in the Swiss

Alps (e.g. Huss et al., 2010; Figure 11). The drop in mean annual as well as summer temperatures in the beginning of the 1970s (Figure 14), combined with a positive spring precipitation anomaly especially in the 1980s (Schmidli, 2000) resulted in several consecutive years with positive mass balance of Alpine glaciers, which has strongly decreased again in recent decades (Huss et al., 2010; Escher-Vetter et al., 2009; Bauder et al., 2007; Vincent, 2002). Zmuttgletscher has shown a similar behaviour, pointing out that its overall magnitude of mass changes has foremost been governed by climatic changes, rather than the debris

cover.

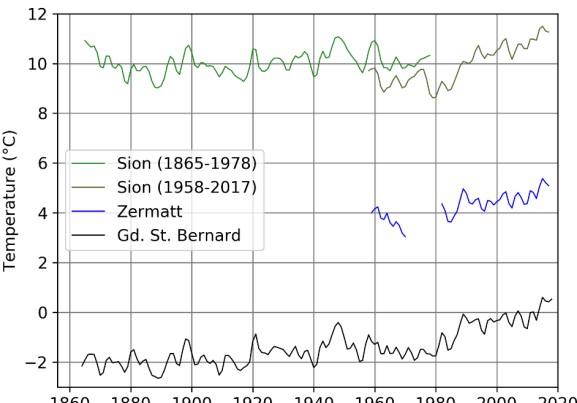

**Figure 14: Temperature time series from Sion, Zermatt, and Col du Grand St. Bernard. The time series of Sion was continued with a new station in the vicinity since the 1960s. Col du Grand St. Bernard is on the same elevation as the glacier tongue (2472 m), but further away (~40 km). The data is smoothed with a five-year running mean.**

As an effect of the warming atmosphere since the mid-19th century, the mass balance of Zmuttgletscher turned negative, first only slightly (1879-1946: -0.09 ±0.12 m.w.e./yr), then more strongly after 1946 (-0.67 ±0.16 m.w.e./yr; Figure 11), leading to glacier retreat and shrinking. On the other hand, the debris-covered area constantly increased from 2.8 ±0.2 km² in 1859 to 3.8 ±0.01 km² in 1961. The temperature rise is probably the main reason for the strong increase of debris-covered area: higher temperatures lead to a rise of the rain-snow transition altitude as well as to more melt during summer and autumn. Therefore,

debris emergence can happen faster and further up-glacier (Stokes et al., 2007). Decreasing ice flow also supports the development of a continuous debris cover of substantial thickness: the emerging debris is evacuated more slowly, thus the debris has more time to melt out of the ice and can thicken. In the mid-20th century a continuous debris cover existed on the lower part of the Zmuttgletscher tongue, likely already responsible for a flattening of the mass balance gradient. Exposed-ice areas and surface flow channels existed in the lowest, debris-covered part of Zmuttgletscher already in 1930 (Figure S13) and

thereafter increased in number and area until 1977. These surface features are partly responsible for elevation change rates that are higher in these areas close to the terminus than further up-glacier.





Subsequently, atmospheric cooling and increased precipitation brought a shift towards less negative mass balance from 1977-1988 (-0.27 ±0.18 m w.e./yr). This is reflected in a slight thickening of the tongue (Figure 5) as well as increased ice flow (Figure 11), leading also to enhanced crevassing in the regions of extensional ice flow. The total debris-covered area during this period was stable but its upper margin was even moved down-glacier in the central glacier parts where flow velocities

increased (Figure 15). The upper margin of the zone with most exposed-ice areas was also moved down-glacier during this period and the total area of such ice cliffs dropped by a factor of 2 (Table 3). The reaction of glacier length and area was less direct, but the retreat and area loss slowly attenuated.

After 1988 mass balances turned again more negative, and flow velocities quickly dropped all over the tongue, first higher up, then also in the lowest regions (Figure 11 and Figure 15b). At the same time, the debris-covered area again strongly increased

– especially during the 1990s – while the glacier area was stable and the central part of the terminus even advanced for some metres, leaving behind a small moraine after starting to retreat in 2001 (Figure S 15). The effect of the preceding period of more positive mass balance is visible in the elevation change patterns of the tongue until the period 1995-2001, when the rate of change was still close to zero at the terminus, whereas it was already highly negative on the rest of the tongue (Figure 9).

During the last 1.5 decades (2001-2017) mass balances have continued on a strongly negative level (~0.8-0.98 m w.e./yr), and

thinning rates have been constantly high (> 1-2 m/yr) all over the tongue. Thinning has been most pronounced in debris-free areas – especially on the tongue of STG – and in areas of exposed ice within the debris-covered part. These thinning patterns have affected length and area changes such that they have been comparably small given the high mass loss (Figure 3 and Figure 11). Flow velocities have continuously decreased after 1988, and since ~2001 the lowest ~1.5 km of the tongue are quasi-stagnant (mostly below 2 m/yr). On large debris-covered glaciers, smaller flow velocities are often the result of a

decreased driving stress due to flat tongues that result from sustained reversal of the mass balance gradient (Bolch et al., 2008a; Anderson and Anderson, 2016). At Zmuttgletscher, the slope has slightly increased over time, therefore we ascribe the decrease of dynamic activity to the reduction in ice thickness and, hence, mass flux and climate, analogue to findings by Dehecq et al. (2019). The main reason for the observed thinning patterns is probably the increasingly extensive and likely also thicker debris cover that is, however, heterogeneously distributed over the tongue (Figure 7). Due to the stagnant terminus, debris can leave

the glacier surface only by terminal retreat or fluvial transport. Decreasing flow dynamics and continuously high ablation will lead to further debris melt-out at the surface, which will increase the debris thickness and in turn decrease ablation. However, in the lowest glacier sections this process is disturbed by surface meltwater channels and ice cliffs and related debris redistribution.



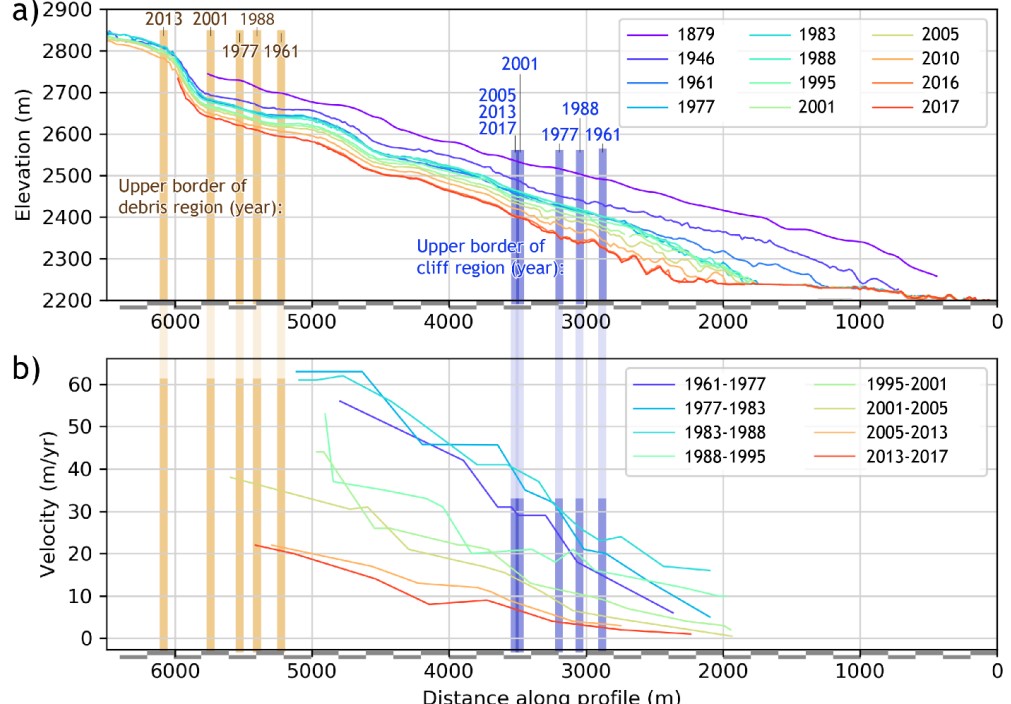

**Figure 15: a) Central elevation profiles over the glacier tongue from the 19th century until today; today´s line (red) marks the former glacier bed from 0 to ~2000 m along the profile, the rugged lower part of the lines is shaped by ice cliffs and flow channels. b) Surface flow velocities from the 1960s until today along the same profile; velocities were highest during the positive mass balance period in the 1970s/1980s; the lowest part of the tongue is almost stagnant today. Vertical brown and blue bars show the upper borders of the debris-covered area and the region of ice cliffs, respectively; both borders have interrupted their up-glacier trend between 1977 and 1988.**

#### 5.2.2 Debris-related surface features on Zmuttgletscher

Judging from field visits and DEM differencing, the most relevant surface features leading to exposed-ice areas such as ice cliffs – and hence enhanced ablation – on Zmuttgletscher are supraglacial meltwater channels. Surface melt is often running off superficially over long distances and several water flow channels exist all over the glacier tongue, outside of crevassed areas. Inside and alongside these channels bare ice often becomes exposed when water washes away the debris or laterally cuts into the ice (Figure 16a), or because locally differential melting steepens the channel walls until the debris slides off. The location of supraglacial meltwater channels on Zmuttgletscher seems closely related to areas of compressional flow – in flat areas and troughs between ridges – and has been most pronounced in the lowest ~1.5 km, where also most exposed-ice areas (mostly ice cliffs) form. The up-glacier migration of the upper margin of this zone stopped in 2005, just below a small section of steeper surface slope with extensional flow (Figure 15). Exposed-ice areas also develop when en- and subglacial cavities – either slowly or rapidly – collapse and the shear planes of the fractured ice get exposed (Figure 16b). We suppose that abundant englacial ablation is effectively creating and enlarging such cavities (Immerzeel et al., 2014; Fischer 2015; Figure 16c) on Zmuttgletscher. Supraglacial ponds are known to have a high potential to increase ablation (Sakai et al., 2000; Röhl, 2008; Miles et al., 2016) and are also potential origins of ice cliffs (Figure 16d), but their relevance on Zmuttgletscher is low due to their small number and area.





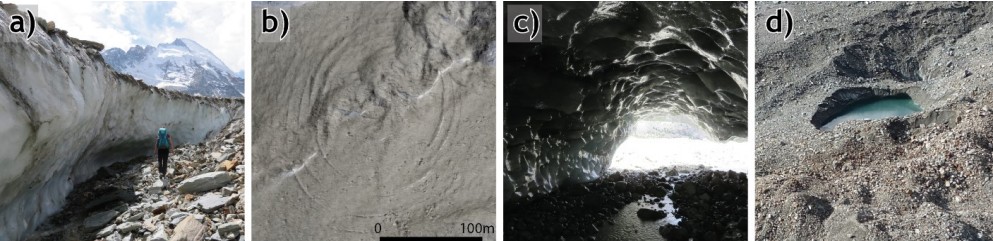

**Figure 16: Examples of superficial and englacial ablation and their consequences. a: supraglacial pond and ambient cliffs. b: subglacial water channel. c: meander of supraglacial water channel with confining, under-cut ice cliff. d: almost circular surface lowering as a consequence of en- and subglacial ablation.**

Large terminal ice cliffs also exist at the glacier mouth(s) and are responsible for a fast retreat in these locations compared to where the terminus flattens out and the debris stays on the ice. The situation of a terminal cliff at the glacier mouth combined with terminus retreat is also found e.g. at Gangotri glacier (Bhambri et al., 2012; Bhattacharya et al., 2016), whereas some glaciers with a stable terminus (e.g. Khumbu, Miage) do not show such terminal cliffs (Bolch et al., 2008a; Diolaiuti et al., 2009). Remarkably, depressions and features of irregular surface topography near the terminus are already indicated in the

Siegfried map from 1879. Certain non-terminal ice cliffs on Zmuttgletscher have reached >25 m in height and have persisted for over two decades. Consistent with the general literature, the consequences of areas of exposed ice at Zmuttgletscher are

-   a more chaotic pattern of surface elevation changes due to locally strong ablation,
-   a strong redistribution of debris through cliff backwasting and sediment evacuation and deposition by water (this redistribution process is difficult to determine but can potentially be important because it leads to a more

15       heterogeneous debris cover),
-   and stronger surface elevation changes at the lower part of the tongue (especially downstream of a small topographic step, see Figure 15a at profile distance ~3500 m). As a result, the elevation change is still stronger towards the terminus, which would not be the case without these exposed-ice areas.

Nevertheless, the exposed-ice areas are only to a small extent responsible for the high volume loss on a glacier-wide scale (in

the range of ~5%) because the total exposed-ice area is small (Table 2). On the one hand, exposed-ice areas are smaller compared to other glaciers, e.g. in the Himalaya, where much higher contributions of cliff melt to the total ablation have been observed (e.g. Sakai et al., 2000; Ragettli et al., 2016; Juen et al., 2014). On the other hand, the thinner debris on Zmuttgletscher as well as the conservative calculation method (including buffer areas around exposed-ice areas) might partly explain the smaller elevation change contrast between areas with and areas without exposed ice. The upper boundary of exposed-ice areas

has been stable for the recent two decades, presumably due to the steepening in surface profile and higher flow velocities just up-glacier (Figure 15). Even though high flow velocities may be expected to subdue the emergence of exposed-ice areas, we could not find a clear velocity threshold linked to their occurrence, e.g. to designate areas of potential ice cliff formation. Interestingly, during the period of more positive surface mass balance in the 1970s and 1980s, the glacier became more dynamic, and the upper boundary of exposed-ice areas moved downstream (Figure 15). Further, during this period the absolute

and relative area of exposed ice was clearly reduced compared to both before and after (Table 3). This suggests a clear link between dynamic state and the occurrence of ice cliffs and would imply expanding exposed-ice areas on stagnating tongues, consistent with the general interpretation of observations (Pellicciotti et al., 2015).

### 5.3    Placing of results in a wider context

Length and volume and changes of glaciers for a given climate history are strongly dependent on glacier geometry and, thus,

individual response time scales (Jóhannesson et al., 1989). Therefore, one has to be careful in comparing the length change curve of Zmuttgletscher with other glaciers. However, such regional comparisons still provide some useful context. The retreat of Zmuttgletscher is relatively modest compared to that of other large Swiss glaciers (Figure 3); it has also shown little terminus





fluctuation, even during the climatically favourable period in the 1970s and 1980s. Other glaciers with similarly subdued fluctuation are Unteraargletscher and Glacier de Zinal, both of which are also debris-covered in their lower reaches. Nevertheless, their retreat since the 1880s has with 1.7 and 2.5 km (i.e. -21 m/yr and -14 m/yr) been considerably stronger, albeit a similar total length (~10 and ~7 km today, respectively) as Zmuttgletscher (~7 km; GLAMOS, 2018). Unlike smaller

and debris-free glaciers, neither Unteraargletscher nor Glacier de Zinal advanced in the 1980s and 1990s (Figure 3). Aletschgletscher is too large and thick to react to short periods of positive mass balance and shows a clear, but increasing long-term retreat trend since the end of the LIA (i.e. -24 m/yr). On the other hand, Findelgletscher, a nearby debris-free glacier of similar size and thickness as Zmuttgletscher and stretching over a similar elevation range, showed a sharp advance in the 1980s (1979-1985: +246 m, i.e. 41 m/yr), followed by a strong retreat since then (1985-2016: -1355 m, i.e. -44 m/yr; in comparison

Zmuttgletscher 1983-2016: -7.2 ±0.01 m/yr). Also during periods of strongly negative mass balance at Zmuttgletscher, its retreat rate has been smaller than that of many other glaciers. Area changes from the beginning of the 1970s until ~2012 for the same sample reveal that Zmuttgletscher has preserved 93% of its area since then (i.e. -0.17 %/yr), which is comparatively high (Figure 3b). During a similar period, Findelgletscher has preserved only ~88.6% of its area (i.e. -0.32%/yr). Thus, in terms of length and area changes, Zmuttgletscher shows a distinctly delayed reaction to climatic perturbations compared to

debris-free glaciers in the region.

Comparing Zmuttgletscher to observations from Himalayan glaciers leads to no coherent message about the influence of debris cover on glacier evolution, because of the heterogeneous pattern of glacier length and area evolution. Some Himalayan glaciers have changed more strongly than Zmuttgletscher, e.g. Gangotri glacier (Bhambri et al., 2012; Bhattacharya et al., 2016) and glaciers in North-Western India (Kulkarni et al., 2011; including debris-free glaciers). Most studies, however, yield smaller

numbers, which range in the dimension of Zmuttgletscher (e.g. area changes in the Sagarmatha region; Salerno et al., 2008; Bolch et al., 2008a) or even much below, like in Langtang (Kappenberger et al., 1993) and Bhutan (Karma et al., 2003). Compared to the evolution of Swiss glaciers, these numbers do not allow a clear distinction between the reaction of debris-covered and debris-free glaciers. To better understand this reaction and ease a direct comparison, longer histories of changes of geometry, climate, and debris cover of individual glaciers need to be considered to account for long response times.

The effect of relatively uniform thinning over much of Zmuttgletscher's tongue can be attributed to the debris cover and is observed at several debris-covered glaciers in Switzerland. By dividing a sample of Swiss glaciers into ten vertically equal elevation bins, the thickness evolution of these glaciers of various sizes, elevations, and elevation ranges can be compared. For the period 1980-2010 debris-covered glaciers including Zmuttgletscher (Figure 17) showed smaller or constant elevation changes in the lowest elevation bins than higher up. In contrast, debris-free glaciers mostly yielded an increase in elevation

change rates until the lowest elevation class. In case of similar geometries of the glacier tongue, such a pattern will lead to a slower adaptation of glacier length and area of the debris-covered glaciers and hence more extended tongues, at least until most of the ice has depleted or the tongue gets separated from the active glacier part at a topographic step with thin ice.





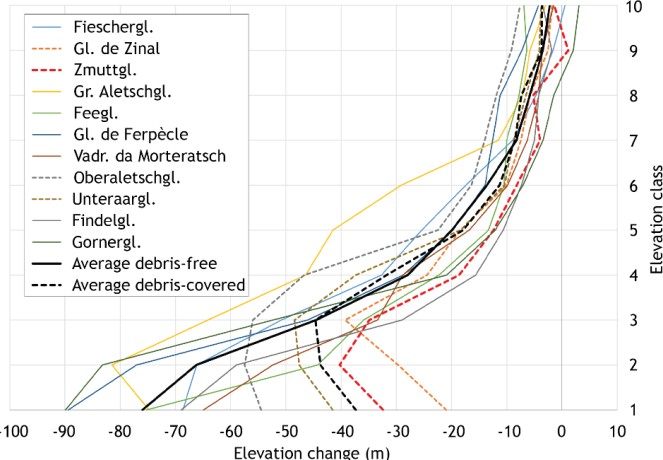

**Figure 17: Normalised elevation change betwee 1980 and 2010 for a selection of Swiss glaciers. Solid lines: mostly debris-free glaciers, dashed lines: debris-covered glaciers. Data from Fischer et al. (2015). The lowest section of Aletschgletscher (yellow line) is actually also debris-covered.**

A similar flattening or even reversing of the elevation change gradient over parts of the glacier tongue is a typical feature of debris-covered glaciers, which has been observed also in other regions, e.g. in the central Himalaya (Inoue, 1977; Bolch et al., 2008a; Benn et al., 2012; Ye et al., 2015; Ragettli et al., 2016) and the Tien Shan (Pieczonka and Bolch, 2015).

The mass balance of Zmuttgletscher from 1879-2017 (-0.31 ±0.03 m w.e./yr) is in the range of many other Swiss glacier's mass balance (Bauder et al., 2007). Thus, unlike what Thomson et al. (2000) found for Glacier de Miage in Italy (positive mass

balance on the tongue over the 20$^{th}$ century), the debris cover on Zmuttgletscher does not seem to have a strong influence on the century-long mass change. During the 1970s/1980s, Zmuttgletscher's mass balance was comparable (only slightly negative or even positive) to other glaciers in Alps, and also the strongly negative mass balance of close to -1 m w.e./yr since ~2001 is observed at other large Alpine glaciers (e.g. Findelgletscher, Glacier de Zinal, Glacier d'Argentière; World Glacier Monitoring Service, 2017). Therefore, the century-long and also the decadal reaction of mass change at Zmuttgletscher is comparable to

debris-free glaciers in the Alps. In the second half of the 20$^{th}$ century, the mass balance of Zmuttgletscher (-0.42 ±0.19 m w.e. for 1961-2001) was slightly more negative than most values from the Himalaya, which generally range between -0.1 and -0.35 m w.e. (Dobhal et al., 2008; Bolch et al., 2012; Zhou et al., 2018; Azam et al., 2018). Also in central Tien Shan glaciers were found to lose mass at similar rates for the period 1976-2009 (ca. -0.27–0.43 m w.e./yr; Pieczonka et al., 2013).. A major reason for the often much lower mass balance values of Zmuttgletscher compared to the Himalaya is presumably the thicker

debris layer on Himalayan glaciers exerting a stronger insulation effect (>0.5 m; Patel et al., 2016; Rounce and McKinney, 2014; McCarthy et al., 2017).

Stagnant glacier tongues have often been observed at strongly debris-covered glaciers (e.g. Bolch et al., 2012), which is also true for Zmuttgletscher today. The main reason for the slow-down is the decrease in ice flux, rather than the long-term effect of debris cover leading to a flatter glacier tongue. Positive mass balances directly impact flow velocities, which was observed

in the 1970s and 1980s at Zmuttgletscher, but also at debris-covered Glacier de Tsarmine (Capt et al., 2016) and a number of other debris-free glaciers in Switzerland (Glacier du Giétro, Glacier de Corbassière, Mattmark; Bauder, 2017) and Austria (Hintereisferner, Kesselwandferner; Stocker-Waldhuber et al., 2018). Also at Glacier de Miage, a kinematic wave migrating down-glacier was observed between the 1960s and 1980s that led to a small terminus advance in the late 1980s (Thomson et al., 2000). Even the long-term (1960s until today) reduction in flow velocities on the tongue of Zmuttgletscher is in line with

observations from all of the above mentioned glaciers. All these examples show a direct reaction of flow velocites to climatic changes regardless of debris coverage. However, the extensive debris cover slows down the terminus retreat of Zmuttgletscher and is thus – indirectly – a contributary cause of very low velocities on the lower part of the tongue. Zmuttgletscher's debris



extent has increased by ~19% points since the end of the LIA without evidence of particularly large rock falls. The increase has happened especially by widening and up-glacier expansion of existing longitudinal debris features, along and next to those glacier regions that are partly fed by avalanches. Potentially relevant amounts of debris input originate from lateral moraines that border Zmuttgletscher and its SBG branch in the lower region of the tongue, but their influence is likely constrained to

the glacier margin (van Woerkom et al., 2018). Changes – mainly increase – of debris-covered area have been shown for other Alpine glaciers, the Himalaya, and the Caucasus, but have so far not been quantified for a century-long period: Deline (2005) presented a qualitative description of supraglacial debris evolution for several glaciers in the Mont Blanc region since the 18$^{th}$ century; Kellerer-Pirklbauer et al. (2008) showed that the debris-covered area of Pasterze in Austria increased from 5.4% to 7.3% between 1964 and 2000; Stokes et al. (2007) found an increase of 3-6% points at various glaciers in the Russian Caucasus;

Gibson et al. (2017) found a slight decrease of debris-covered area on Baltoro glacier from 2001-2012; Bhambri et al. (2011) found an increase of debris-covered area in the upper Gangotri basin but didn't quantify it. Kamp et al. (2011) documented an increase of 10.4 and 7.9 percentage points for two glaciers in Ladakh between 1990 and 2006. The longest time period of quantified debris cover change exists for Khumbu Himal, for which Bolch et al. (2008a) quantified an increase from 39% to 42% between 1962 and 2005, thereby confirming an earlier analysis by Iwata et al. (2000). The values above suggest that the

debris cover increase at Zmuttgletscher has been comparatively strong. Main reasons are a combined effect of elevated temperatures and decreasing velocities, the large debris-free ablation area that still existed a few decades ago, and likely also the comparatively small glacier size, i.e. the shorter response time. Similar to Zmuttgletscher in the 1970s/1980s, Deline (2005) documented a decrease in debris-covered area for Glacier de Miage between ~1890 and ~1920, caused by an increase in surface elevation and flow velocity (Thomson et al., 2000). Thus, changes in glacier dynamics can directly impact supraglacial debris

extent at the transition between debris-free and debris-covered glacier surface. The debris thicknesses we found (mostly 10-25 cm) were similar to other Alpine glaciers, e.g. Unteraargletscher (Bauder, 2001; Sugiyama, 2003) or large parts of Glacier de Miage (Mihalcea al 2008; Foster et al., 2012), but smaller than on many glaciers in the Himalaya and Karakoram with average values of >0.5 m (Nakawo et al., 1986; Patel et al., 2016; Rounce and McKinney, 2014; McCarthy et al., 2017).

## 6    Conclusions

This study presents an over 150 year-long reconstruction of glacier geometry, surface topography, and debris cover from the LIA until today, as well as over five decades of surface flow velocities. Debris cover extent has more strongly increased than shown elsewhere (from ~13% of the total glacier surface in 1859 to >32% in 2013). Despite the expansive debris cover, we conclude that the evolution of Zmuttgletscher has been mainly driven by climatic changes, both on a decadal and on a century-long scale. The heterogeneous debris thickness distribution reduces ablation from >15% to up to 80% (compared to bare ice)

over large parts of the debris-covered area. Even if this debris cover is relatively thin (mostly ~15 cm), it has efficiently attenuated and delayed the reaction to climatic changes in terms of area and length adaptations compared to other glaciers in the Alps, both during periods of mass gain and mass loss (overall values of -12.1 ±0.09 m/yr and -0.17 ±0.02% since ~1859, respectively). In contrast, rates of surface elevation change and glacier-wide mass balance are comparable to other debris-covered, but also debris-free Swiss glaciers. Nevertheless, debris cover affects the spatial variability of elevation changes: (1)

On a tongue-wide scale, the gradient of elevation changes is (almost) constant, as is typical for many debris-covered glaciers. The major reason preventing a more pronounced/inversed gradient is the presence of exposed ice between the terminus and about 1.5 km up-glacier, leading to enhanced ablation in these areas. Combined with the topographic setting this has led to a slightly increased slope of the glacier tongue, notwithstanding the stagnation of the lower part of the tongue since 2001. (2) On a more local scale, elevation changes are driven by variations in debris extent and thickness, and the emergence of exposed

ice (either originating from channelled surface water flow, or – less important here – from ponding and subaquatic ablation), or englacial ablation and subsequent cavity collapse. Surface flow velocities have approximately halved between the two



periods 2005-2007 and 2016-2017, and today the lowest ~1.5 km of the tongue are quasi-stagnant. Overall, flow velocities have strongly decreased compared to values before and after a peak in the 1970s and 1980s – similar to other glaciers in the Alps – even though the glacier tongue has steepened over time. Thus, the low flow velocities are mainly due to reduced ice flux and only to a minor degree influenced by debris cover. Higher flow velocities between 1977 and 1988 were triggered by increased mass flux from the accumulation area, which also caused local glacier re-thickening. This increase led to a local down-glacier migration of the debris-cover margin, followed by a strong debris-cover increase when post-1988 velocities dropped and temperature rise led to increasingly negative mass balance. These observations proof a strong and direct impact of flow dynamics on debris cover extent. Higher flow velocities also moved the upper margin of areas of exposed ice (mostly ice cliffs) down-glacier, temporarily reduced the area of exposed ice, and eventually led to a slight advance in the 1990s. These findings suggest a clear influence of flow dynamics through the enhancement of ice cliff formation in stagnating glacier parts and vice versa. The above described processes and feedbacks are likely valid and relevant for other debris-covered glaciers in the Alps and elsewhere at potentially different rates and magnitudes. In the context of global warming, it is therefore crucial to include these findings in models for glacier projections.

## 7    Acknowledgements

This study was funded by the SNSF project #200021_169775 "Understanding and quantifying the transient dynamics and evolution of debris-covered glaciers". We thank M. Fischer and H. Machguth for making the elevation change dataset over all Swiss glaciers available. Thanks to R. Ganarin for conducting and supporting debris thickness measurements. Thanks to the 3G group for the constructive discussions.

## 8    Author contributions:

N. Mölg, T. Bolch and A. Vieli designed the manuscript. N. Mölg generated the DTMs and Orthophotos and performed all analysis. A. Walter performed radar measurements and processing and provided the respective text section. N. Mölg wrote the draft of the manuscript. All authors contributed to the final version of the manuscript.

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
