# Peer review of "Unravelling the evolution of Zmuttgletscher and its debris cover since the end of the Little Ice Age"

_The Cryosphere, 2018_

## Referee Comment (RC1) · Rounce (Referee) · 1 Feb 2019

**Review of "The role of debris cover in the evolution of Zmuttgletscher, Switzerland, since the end of the Little Ice Age" by Mölg et al.**

The study describes the evolution of Zmuttgletscher from the Little Ice Age (~1850) to present combining DEMs, aerial/satellite imagery, and field work. The study finds that the glacier's evolution is mainly driven by climatic forcing, and that the debris cover impacts how the glacier evolves in response to these forcings. Specifically, the debris cover reduces the rate of length and area changes compared to clean ice glaciers as it causes the glacier to development a low-sloped, stagnant tongue as opposed to simply retreating upglacier. The authors make the distinction clear however that this gently-sloped tongue is caused by a change in the ice flux upglacier as opposed to any differential melting caused by changes in the spatial variability of debris thickness over a long period of time.

I found one of the most impressive elements of this paper to also be the cause of my biggest critique, i.e., the study incorporates a broad range of data/methods (generation of historical DEMs, geodetic mass balance estimates, surface velocity measurements, debris thickness field surveys, ablation stakes, etc.), which provide a great deal of information to evaluate the evolution of this glacier, but also make this paper very long and at times sound repetitive. This is clear in that there are 17 figures in this paper, many of which report the same data in a different format. For example, Figure 17 shows the normalized elevation change, which is similar to Figure 10a (elevation change vs glacier section), which is similar to Figure 9 (spatial map of elevation change). The changes in area, debris-covered area, and length are shown in Figures 3, 4, and 5, yet they are all linked together (and Figure 3 has a table of the values). The influence of ice cliffs is shown in both a table (Table 3) and a figure (Figure 8). Similarly, portions of the text are repetitive, e.g., elevation change vs. debris thickness is shown as the elevation change, the thinning rate, and then as the ratio between the two, so the same values are reported 3 separate times in a row. I encourage the authors to get creative in how they present this data, and think about what data is critical to show in a figure or table, and what can be moved to the supplementary material. If these changes are made, then I think the text will also become more concise, and the conclusions of the study will become more prominent.

Overall though, I think this is an important contribution to the literature and advances our understanding of the evolution of debris-covered glaciers. References to previous studies were very thorough and results were placed in broad context (perhaps too broad in some cases). There were many grammatical errors throughout; however, these grammatical errors are minor revisions. That said, while I believe the methods are rigorous, the amount of data is impressive, and the conclusions are well supported, all of which would justify the manuscript to be accepted, I think it requires fairly significant revisions to reduce the repetitiveness of the figures/tables and text. Therefore, I recommend the manuscript be reconsidered after major revisions.

General Comments
Section 4.4.3 – Given that this is describing the mass balance, i.e., the combination of the surface mass balance and the flux divergence, I would suggest placing it after you have discussed the changes in surface velocities. For example, "the wave of additional mass flux" has no meaning here given that surface velocities have yet to be mentioned.

There are multiple figures and tables that appear to show the same thing. This is repetitive (e.g., Table 3 and Figure 11). I commend the authors for explicitly stating the data such that it can be used by other studies in the future; however, if it is simply taking up space in the manuscript and repeating information that is shown visually, then it should be provided in the supplementary material. Additionally, there are many figures that have tables included within them, which seems unusual (e.g., Figures 3 and 10). The manuscript is already quite long with 17 figures and 3 tables, so this could help to make the study more concise.

Specific Comments
P refers to page
L refers to line number
*Italics* indicate suggested grammatical changes

P1 L2 – changes *in* debris cover over time

P1 L2 – "changes in debris cover over time, or surface flow velocities" is a bit unclear. Do you mean they are investigating changes in debris cover or surface flow velocities over time? Or is this surface flow velocities meant to be on its own so there are 4 separate items here? Please clarify.

P2 L7-12 – This paragraph is a single sentence. Consider breaking this paragraph down into multiple sentences to make it easier to read. Also, are referring to all the studies that have been conducted on debris-covered glaciers? Or are you referring to the numerical modeling studies?

P2 L18 – increase *in* debris cover. General note as this is the second time, increase "of" typically followed by a number, while increase "in" typically refers to the object/thing. Consider revising this change from "of" to "in" throughout the text.

P2 L22-23 – if you are going to say they are in the center of attention due to their importance, then you should state the reason why they are important. I'm not sure this is necessary though, since the main point is simply that the data is not available to investigate.

P2 L28 – "at the example of" doesn't make sense. Consider "… debris-covered glaciers through the study of Zmuttgletscher in the Swiss Alps."

P4 L4-7 – Remove from the caption of Figure 1.

P4 L10 – Table 1 shows the topographic maps, satellite, and aerial images, so move "(Table 1)" to after the satellite images. You could consider breaking this into two separate blocks of data like "topographical maps, …, and satellite images (Table 1) in addition to various field observations and long-term temperature measurements." To make the distinction that Table 1 is related to these 3 products even more apparent.

P5 L6 – First time DTM abbreviation is used it should be spelled out fully.

P6 L15 – "This resulted in a glacier area information for each data" doesn't make sense. Please clarify. Also, is this different than the first line of this section stating that glacier area was measured since 1859? It appears to be repetitive.

P6 L24 – until 1997, *and* an additional data point…

P6 L32-33 – A bit unclear: were the two historic maps with debris cover symbols manually digitized as well? If so consider, "Debris cover extent for the orthophotos and historic maps that contained a debris cover symbol were manually digitized (Figure 2a and b)."

P6 L33 – … Siegfriend map (1879) *was* verified using two photographs…

P6 L34 – This information *was* valuable …, which *was* the region of the strongest changes. Switching from past tense in first sentence to present tense in this last sentence. Suggest keeping the tenses the same for the paragraph to make it easier to read.

P7 L5 – A bit confusing. First, you should reference Figure 1, since they show where these were provided. Second, what do you mean by "for setting the elevation change observations into context"? Do you mean you were doing this to compare to the elevation change estimates from aerial/satellite imagery? Or were you measuring the surface mass balance to be able to break the elevation change in the aerial/satellite imagery into the surface mass balance and the flux divergence? I assume this will become clearer in the results, but it should be clarified here as well.

P7 L12-14 – A bit unclear as well. Do you mean "Debris thickness was measured via manual excavation along several transects perpendicular to the glacier tongue."? This seems to be the case from the figure and results.

P7 L15-16 – "used to put observations into context" does not provide any information to the reader. What observations were you trying to put into context? What context? Additionally, stations had to be close to what? To each other? To the ablation stakes? Please clarify.

P8 L28 – *which* reaches up to …

P8 L40-41 – consider consolidating the two sentences "Independent of the method, 100 m elevations bins (starting from 2100 m) were used to get a representative value for elevation change calculations in order to reduce the susceptibility of elevation change calculations to outliers due to the incomplete coverage of some areas of the glacier."

P9 L22 – Do you mean, "change in debris thickness may impact the changes in surface elevation"?

P9 L25 – 20,000 m$^2$? Or 20.0 m$^2$?

P9 L37 – delete "additionally"

P10 L2 – The interferograms *were*…

P10 L4 – negative velocities *were* considered *to be* noise…

P10 L18 – this *study*.

P10 L11 – The wording is a little awkward.  Consider "From close to the end of the LIA (~1850s) to 2017, Zmuttgletscher has retreated by 1907 +/-12 m (12.1 +/- 0.09 m/yr).  The maximum rate of retreat peaked between 1961 and 1977 at 21.7 +/- 0.04 m/yr."

Figure 3 – is there a reason Figure 3b, which is a table, is listed as a figure?  What is the cause of the difference between Zmuttgletscher (this study) and GLAMOS 2018?  They appear to be substantially different, albeit showing similar trends.

P10 L25 – "was with" does not make sense.  "Until 2013" also sounds awkward.  Consider something along the lines of "At the end of the LIA 2.8 +/- 0.2km$^2$ of Zmuttgletscher was debris-covered, which has increased to > 5 km$^2$ in 2013.  During this time, the total glacier area has …"

P11 L4 – "all parts of the glacier" is fairly vague, perhaps in "all tributaries"?

P11 L5 –  the *extent of* debris has expanded …

P11 L5 – you refer to the debris-covered extent getting closer to Dent d'Herens, but isn't SBG (which is included in since you state "both" at the beginning of the sentence) getting equally as close to the headwalls of Dent Blanche?

P11 L4-10 – I think it's important to be specific when referring to properties of the debris cover.  In this paragraph you are discussing the extent of debris cover, so this should be made clear.  The "debris cover grew strongly" could refer to the thickness of debris cover or the extent of debris cover.  Since both are discussed in this paper, it's important to always distinguish between the two.

P11 L8-10 – This sentence is very confusing.  What does "after the exposure of the rock wall" mean?  Please clarify.

P11 L18-20: difficult to read.  Consider "In such cases, the typical base layer consists of fine-grained material (sand and even silt) lying directly on the ice, which is overlain by a few centimeters of pebbles.  Above this typical base layer, there is no specific sorting."

P11 L12 – You state these are over several periods, but the caption states these measurements are from 05/07 – 22/08 2017, which is a single period in time.  Please clarify.

P12 L20 – what does "number" refer to?  The number of ice cliffs?

P12 L21 – "lower parts of the glacier tongue until the terminus" is redundant.

P13 L2 – delete "only very"

P13 L10 – delete "only"

Figure 8 – there appears to be a line along the top of the figure?

P13 L19-20 – A bit redundant saying in some periods, then stating the period, and then doing the same later. Additionally, I'm not sure "strong" is the best choice of words to describe elevation change. It'd be better to be specific: "most negative" or the "surface lowering was greatest", something along those lines. For example, "The elevation change was most negative near the terminus between 1946-1977. Since 2001 though, the elevation change over the tongue has become more heterogeneous."

P13 L22 – "Between 2879 and 1946, the *average* surface elevation change was -0.63 m/yr and most pronounced at the terminus. *Surface lowering increased* to …". Note: This is a bit repetitive of L19-20.

P13 L26 – the tongue's *average* surface elevation *change* was almost balanced at -0.11 m/yr.

P14 L1 – use either "surface lowering" or "surface elevation change". Previously, you've used "surface elevation change", so be consistent. This is especially true because you're using positive and negative values for elevation change, so surface lowering with a positive value makes this difficult to read.

P14 L3 – *slight*

P16 L4 – do you mean the ratio of thinning for thick debris compared to bare ice?

P16 L14 – please clarify what "no clear hint of an increasing balancing effect" means? There is no clear trend in the mass balance gradient flattening towards the terminus?

P16 L24 – again, this appears to be a ratio. Ratios are not very intuitive. For example, if both regions were positive, they could still have the same ratio. Why are you reporting and discussing the ratio and changes in ratios? Is it providing information about how "connected" the upper and lower portions of the glaciers are? Is it used to understanding the dynamical state that the glacier is in? Please add a sentence or two to clarify what the ratio is actually telling us about the glacier.

P16 L26 – *still show*

Section 4.5 – Is this referring to the "glacier-wide" mass balance? If so change the heading.

P17 L1 – showed an *increase in elevation* …

P17 L2 – After 1988, the mass balance *became* more negative again, even *while* the lowest areas on the tongue *had a* stable or positive mass balance.

P17 L4 – When did this negative trend intensify? Rearrange this sentence if it was in 2001.

P18 L4 – add the year that this was done again for reference, so that it is clear in the text. It is obvious this this is an independent measurement, so you can delete "as an independent measurement"

P19 L13 – and *thus have* little impact …

P19 L14 – delete "also"

P19 L31-L36 – this reads as a methods section, i.e., the reason you chose the methods you did as opposed to a discussion section. Consider moving this to the methods, although I leave this choice up to you. The other ones "suitability" of methods seem to discuss other studies, shortcomings, etc., which is why I don't think the other parts of this subsection are out of place.

P19 L39 – "Nevertheless we think…" are you referring to the long-term trend or are you saying that despite potentially underestimating the area, you still think the trend is significant?

P20 L13-5 – "similar behavior", perhaps similar trend? "pointing out" also sounds awkward here. Perhaps "Zmuttgletscher has shown a similar trend indicating that its glacier-wide mass balance has foremost been governed by climatic changes rather than a response to changes in debris cover."

P20 L31 – Be careful with "higher" elevation change rate as this could imply a positive elevation change, but here it seems to be referring to being more negative? Please clarify.

P21 L3-5 – A bit confused here as to how the debris-covered area was stable if the upper margin of the zone was moved down-glaciers? Are you suggesting that the glacier advanced and that's what kept the debris-covered area constant? Otherwise, if the upper margin, which I assume is referring to the debris/ice extent interface moved downglacier, how could the debris extent remain constant?

P21 L8-10 – "again" typically does not follow the verb, e.g., became more negative again, or the debris-covered area strongly increased again.

P21 L23 – and *also likely* thicker debris

P25 L18 – delete a period

P25 L18-21 – In Section 5.2.1 you state that the magnitude of mass change is governed by changes in climate as opposed to changes in debris cover. However, here you are stating that the different magnitudes in mass balance between Zmuttgletscher compared to glaciers in the Himalaya is likely attributed to the differences in debris cover. While this could play a role in

the differences between the two, how do you know that this difference is not caused by differences in climate forcing?

P26 L6-14 – Can this be consolidated?  While it is nice to place the changes in debris-covered area into a regional context, the point gets a bit lost in this long list of every study, region, and change.

P27 L5 – "re-thickening"?  It thickened, so why include the "re-"?

P27 L7 – These observations *prove* a …

---

## Referee Comment (RC2) · Philip Kraaijenbrink (Referee) · 15 Feb 2019

Review in supplement

Please also note the supplement to this comment:
https://www.the-cryosphere-discuss.net/tc-2018-292/tc-2018-292-RC2-supplement.pdf

---

## Author Comment (AC1) · 1 Apr 2019

General reply

First of all, we thank both reviewers for thoroughly reading through the manuscript and for all the helpful critique, ideas, and suggestions. We greatly appreciate the valuable comments and the promptness of the feedback.

We acknowledge the view of the referees highlighting the long-term evolution aspect, the general relevance, as well as the extensive and unique set of data and analysis.

They also came up with critical comments and major drawbacks, which mainly referred

to the issue of losing the focus on the main message in the number of different analysis and the overall length of the manuscript. We are thankful for these valuable comments and think they will considerably improve the manuscript.

Here we reply to the major concerns raised by the reviewers and present our ideas for revisions and amendments. The mainly encompass the removal of parts of the analysis, a shortening and streamlining of the discussion, the removal of figures, parts of figures, or figures being shifted to the supplements, and the combination of data presented in figures.

The majority of the specific/minor comments refer to language or detailed content or are specific examples of the major concerns. All these corrections will be addressed in the revised manuscript.

We would greatly appreciate the view of the editor on the suggested revisions before our submission of the revised manuscript.

For our reply to the major concerns please look at the supplement pdf.

Please also note the supplement to this comment:
https://www.the-cryosphere-discuss.net/tc-2018-292/tc-2018-292-AC1-supplement.pdf

―――――――――――――――――――――

---

## Editor Comment (EC1) · Etienne Berthier (Editor) · 9 Apr 2019

Dear authors,

Thanks for your initial and general reply to the reviews.

Their comments were partly echoing my access review where I already suggested (i) a reduction in the length of the text and in the number of figures in order to increase the impact of your study and (ii) the need to avoid new figures/results in the discussion. So there is a consensus with the reviewers on these aspects.

I generally agree with your initial responses and look forward to read a revised, shortened article. You made the decision to keep it as a single paper. This is a choice I respect but make sure you convey a clear message to the readers.

As you know, I can only take my decision after reading the revised manuscript and, very likely, consulting the referees. At this stage, I encourage you to take carefully into account their constructive comments.

Looking forward to read your revised manuscript,

Etienne Berthier

---

## Author Response (AR1)

**The role of debris cover in the evolution of Zmuttgletscher, Switzerland, since the end of the Little Ice Age**

by N. Mölg et al.

Submitted for review to *The Cryosphere* in December 2018

**Reply to reviewer's comments**

**Update and adaptation of the reply from April 1st.**

**General reply**

First of all, we thank both reviewers for thoroughly reading through the manuscript and for all the helpful critique, ideas, and suggestions. We greatly appreciate the valuable comments and the promptness of the feedback.

We acknowledge the view of the referees highlighting the long-term evolution aspect, the general relevance, as well as the extensive and unique set of data and analysis.

They also came up with critical comments and major drawbacks, which mainly referred to the issue of losing the focus on the main message in the number of different analysis and the overall length of the manuscript. For the revision of the manuscript we thoroughly considered all the critiques and suggestions by the reviewers. Changes have been implemented in the revised manuscript, which has – as we think – improved substantially.

Following this overall comment, we first reply to the major concerns raised by the reviewers and present the changes that have been performed in the revised manuscript. They encompass a number of additional changes, i.e. text has been rephrased, added and removed (revised manuscript is ~2000 words shorter), restructuring of content and chapters, removal and re-organisation of figures. The changes performed are consistent with the ones we suggested in our first reply. Changes in reaction to specific comments are shown in the respective replies, as is our argumentation where we didn't follow the referee's advice or where we chose a different way of approaching the consideration. For further changes we want to refer the referees to the revised manuscript, which is provided as a track-changed word document and a pdf where all revisions are already included (see supplements zip file).

**Reply to major concerns**

R2: 'The title is a tad misleading'.

We agree with the concern of R2 that the title is not perfectly appropriate for the focus of the study. To better reflect the main content of the manuscript we chose to focus the title on the glacier evolution rather than on debris cover and adjusted the title to:

*'Unravelling the evolution of Zmuttgletscher and its debris cover since the end of the Little Ice Age'*

R1 & R2: 'Text as well as figures are too long and partly repetitive. It is sometimes confusing to follow the story due to the large number of analysis and results'.

We acknowledge the fact that the manuscript was rather long in places and that the presentation of 17 figures significantly contributed to this length. We agree that the main messages of the manuscript got somewhat 'blurred' by the shear amount of data and analysis presented. We also agree that there would be enough information for two stories but the main point of the study is the longer-term glacier evolution and the interaction and feedback between the different variables. We therefore decided not to slice this paper as separating the results would not allow to bring these results together and investigate their inter-relationships (dynamic feedbacks).

Overall, we have made the text more concise, removed three figures completely and additionally three parts of figures, present some of the content in different format and even removed some smaller analysis which are not relevant for the main outcomes. Specific details of the performed changes are listed below:

- Removal of the generation of a moraine-based DTM for the end of the LIA (part of chapter 3.5) as it was not used for further analysis.
- Removal of the elevation change of different debris thickness classes (former chapters 3.6.2 and 4.4.3). The message of this analysis is the insulation effect of the debris cover, which is already contained in the analysis of ablation stakes.
- Shift of Figure 3b (the table) to the supplements. It is not essential in the main text (but still useful to refer to in the discussion).
- Removal of Figure 6. The debris surface is described in the text and the photos have been placed in the supplement.
- Substantial shortening of the chapter on tongue-wide elevation changes (new chapter 4.6), which is now also easier to follow.
- Modification of the concept and figure (Fig. 11) of the mass balance gradient (incorporated into chapter 4.6). As suggested by R2, we have done the analysis for elevation bins of 50 m instead of the glacier sections as in this way it is easier to understand the concept and to present the results. Due to this modification of the concept, do not anymore need the section map (former Fig. 10b) and the table (former Fig. 10c). We have reduced the number of periods from 10 to 7. Also the terminology has been changed from mass balance gradient to elevation change gradient, which is the correct term in this case.
- The temperature time series (former Figure 14) has been integrated into Figure 3 (mass balance) as a small, additional subplot sharing the same x-axis. In this way the evolution of the geodetic mass balance can be directly compared to the temperature evolution and Figure 14 could be removed.
- Former Figure 13 (flow velocities from feature tracking and radar interferometer) has been moved to the supplements. This figure displayed partly redundant information (reduction over time, stagnant frontal part; already contained in Fig. 12) and partly information that is not necessary for the main message.
- Removal of Figure 14 (temperature time series, see comment above on integrating it in Fig. 3).
- Strong reduction of the discussion on the comparison to other glaciers outside of the Alps. Nevertheless, we have kept a basic comparison to give some useful context, e.g. very few studies have compared debris extent changes over time.
- Complete re-organisation of the discussion. It is now structured in the main topics, as they also contain the main messages. The discussion has been shortened by approx. one fourth.

Overall, this resulted in the removal (or shift to supplement) of three figures (Fig.6, Fig.13, Fig.14) and three figure parts (3b, 10b, 10c) as well as the shortening of a significant amount of text (~2000 words).

R1 & R2: 'Text on analysis and results and even the discussion do not always focus on the main messages, making these messages a bit blurry.'

First of all, we have made the main messages stick out clearer in the discussion and conclusion. To make it clear also here, these main messages are:

- Climate is responsible for the glacier-wide mass balance
- The observed debris increase over time is very strong and unprecedented
- Debris cover provokes spatial and temporal change patterns (reduced length and area change)
- Ice cliffs are persistent as features but do not compensate the debris insulation, thus the larger glacier area is responsible for the comparable mass balance
- Ice dynamics directly influences ice cliff as well as debris cover evolution

We have also amended the text in the introduction, results, and discussion to serve more arguments to arrive at these messages. To be clearer about the messages, we have introduced additional sub-chapters in the discussion that specifically contain the main messages. The additional sub-chapters also help to structure the discussion in a more logical way. Together with the language comments by the reviewers as well as an additional correction by a native-speaking glaciologist we are confident that the story is much clearer now, thus increasing the impact of the study.

R2: 'Methods are not always reproducible.'

We further clarified the methods and added additional details. For the technical aspects of the production and detailed analysis of the DTMs (specific comment) we refer to Mölg & Bolch (2017) as they described the methods in detail. Also note that this study focuses on the glacier evolution and the links between variables and is not designed as a technical assessment or development of methods.

Specific steps that have been undertaken regarding this concern are:

- Slight changes in text by adding further details of explanation and amendments of the text for clearer language (e.g. for the geodetic mass balance, debris cover mapping, ice cliff influence)
- Referring to further existing studies where references were not sufficient (e.g. in surface feature mapping)
- Removal of analysis that are not necessary and thereby reducing the complexity of the overall study as well as specific text sections and analysis descriptions (e.g. the moraine-based surface topography)
- Change of analysis to a clearer concept (ice cliff influence)
- Clarification of our terminology to better distinguish between exposed-ice areas and ice cliffs.

R2: 'Ice emergence and horizontal displacements should be considered when calculating the influence of ice cliffs on total elevation change.'

The objective of this analysis is to understand the potential influence of ice cliffs on elevation change. To do so, we compared area-averaged elevation change with or without ice-cliffs and their surrounding areas but could not see substantial differences. Importantly, we do not explicitly compare different time periods. Also, both horizontal displacements and cliff backwasting are inherently accounted for by applying the buffer. Thus, our main conclusions only refer to elevation change per date (not ablation) and are in our view robust and do not require emergence velocities and horizontal displacements. We have adjusted the description to make the objective of the analysis and the conclusion drawn from it clear.

R2: '…The elaborate comparisons of Zmutt with other glaciers could be removed/reduced. … Some of it would be suited for a review on the state of Swiss glaciers.'

We do not quite agree with the referee. A major aspect of this paper is the difference/non-difference in evolution/dynamic behaviour between debris-covered and debris-free glaciers. Thus, we consider it crucial to set the evolution of Zmuttgletscher in context with other clean-ice and debris-covered glaciers in the region (on the basis of simple measures such as length, area, and elevation change patterns).

R2 raised a specific concern regarding the analysis represented by Figure 17. This analysis clearly yields that debris-free and debris-covered glaciers can be separated based on elevation change patterns, and that Zmuttgletscher can clearly be grouped to the latter. The y-axis of the figure shows only elevation bins and no absolute numbers because the absolute elevation was normalised for comparison. This is necessary to account for the different elevation ranges of the glaciers. To make this clear we have changed the axis caption from 'Elevation class' to 'Normalised elevation class'.

R1: 'The influence of ice cliffs is shown in both a table and a figure.'

We decided to keep Table 2 in the main text. It shows information that is complementary to the figure, such as ice-cliff area and area share per date. However, we tried to incorporate it more in the discussion and focus on the cliff area reduction as a consequence of increased dynamics. This is an important result to show as it is related to dynamic interactions (see main messages).

R1: 'I suggest placing [section 4.4.3] after you have discussed the changes in surface velocities.' …

We agree and have adjusted the structure of the manuscript: The surface flow velocities have been moved up and newly represent chapter 3.4 instead of 3.7 in the methods, and chapter 4.4 instead of 4.6 in the results. We follow this suggestion by R1 because the flow dynamics is somehow inherent in the surface features and the mass balance and even referred to in the text.

Unlike announced earlier, the chapter on the elevation change gradient has not been incorporated into the chapter of geodetic mass balances. Because we only considered surface elevation changes the formerly used term 'mass balance gradient' was not correct and we've change it to 'Elevation change gradient'. With this title and the content it is a logical part of the chapter on tongue-wide elevation changes.

The new structure of the manuscript is as follows:

1. Motivation and objectives
2. Study site
3. Data and methods
    3.1. Generation of DTMs and orthophotos
    3.2. Glacier area and length
    3.3. Debris cover, on-site ablation and on-site air temperature
    3.4. Surface flow velocities
    3.5. Surface features
    3.6. Surface elevation changes and geodetic mass balance
        3.6.1. Uncertainties
4. Results
    4.1. Geodetic mass balance
    4.2. Glacier area and length changes
    4.3. Debris cover
        4.3.1. Debris cover evolution
        4.3.2. Debris cover characteristics
    4.4. Surface flow velocities
    4.5. Ice cliffs and related elevation change
    4.6. Tongue-wide surface elevation change and patterns
5. Discussion and interpretation
    5.1. Methods suitability and shortcomings
    5.2. Chronological evolution and process interactions
    5.3. Unprecedented debris cover increase
    5.4. The influence of debris cover on glacier geometry
    5.5. Climate-driven glacier mass balance and flow dynamics
    5.6. The role of debris-related surface features
    5.7. Dynamic interaction between flow velocities, debris cover, and ice cliffs
6. Conclusions

R2: 'To have an accurate curve the melt rates have to be normalized by the clean ice melt rate at the same elevation. Does normalized in this figure mean that you had a stake at the same elevation on a clean part of ice, representatively close to the debris stake, that was used to normalize each point?'

We have changed the depicted values from 'normalised melt' to 'melt per day', since the clean-ice stake used for normalisation was not placed in a comparable climate as the other stakes (which would not be possible). We present the results of the 7-weeks period to proof and roughly quantify the insulation effect of the debris and not to establish an individual 'Östrem-curve' for Zmuttgletscher, which we also do not need for our further analysis. We think that our conclusion of melt reduction can be drawn because (i) debris of the observed thickness does substantially reduce ablation, (ii) the ablation rather depends on debris thickness than on elevation (there is zero correlation between elevation and melt, but an $R^2$ of 0.95 when we apply a logarithmic function to the measurements above 0 cm thickness), (iii) the measurements for two stakes with 5 and 25 cm, respectively, yield almost the same melt rate even though the stakes are located in quite different elevations.

R1: 'Elevation change vs. debris thickness information is repetitive.'

We were not exactly sure what the referee referred to but we reduced the amount of information in the text as well as former Figure 6. Additionally, the analysis of elevation change on different debris thickness classes has been removed as the temporal coverage on debris thickness information is limited.

**Response to reviewer comments**

Reviewer #1

**Specific comments**

P1 L2 – changes *in* debris cover over time
Done

P1 L2 – "changes in debris cover over time, or surface flow velocities" is a bit unclear. Do you mean they are investigating changes in debris cover or surface flow velocities over time? Or is this surface flow velocities meant to be on its own so there are 4 separate items here? Please clarify.
New P2 L6
We've re-ordered the items. It should be clear now: "Current research at many debris-covered glaciers and specifically in the Himalayas is mostly focussed on processes, such as ablation beneath the debris cover and in areas of ice cliffs and ponds, thinning of glacier tongues, surface flow velocities, or changes in debris cover over time."

P2 L7-12 – This paragraph is a single sentence. Consider breaking this paragraph down into multiple sentences to make it easier to read. Also, are referring to all the studies that have been conducted on debris-covered glaciers? Or are you referring to the numerical modeling studies?
The paragraph refers to all studies. We've started a new paragraph so we think this should be clear. We have added a numbering of the different shortcomings to distinguish between the main items of this paragraph (new P2 L12):
"However, most studies … of debris-covered glaciers: (i) time series are often short …; (ii) investigations are often local …; (iii) by considering only one or few variables … ; (iv) studies at glacier scale … ."
As far as we undertstand it, the semicolon can be used to separate independent clauses into sentences, the content of which is closely related. We consider this an element of style and less important for the understanding of the paragraph and therefore prefer to keep it.

P2 L18 – increase *in* debris cover. General note as this is the second time, increase "of" typically followed by a number, while increase "in" typically refers to the object/thing. Consider revising this change from "of" to "in" throughout the text.
Thank you for this tip. We've changed it in several places throughout the document.

P2 L22-23 – if you are going to say they are in the center of attention due to their importance, then you should state the reason why they are important. I'm not sure this is necessary though, since the main point is simply that the data is not available to investigate.
True. The insertion (" – which are in the center of attention due to their importance – ") is not necessary, we've removed it.

P2 L28 – "at the example of" doesn't make sense. Consider "… debris-covered glaciers through the study of Zmuttgletscher in the Swiss Alps."
New P2 L33.
Thank you. We changed it accordingly.

P4 L4-7 – Remove from the caption of Figure 1.
Done.

P4 L10 – Table 1 shows the topographic maps, satellite, and aerial images, so move "(Table 1)" to after the satellite images. You could consider breaking this into two separate blocks of data like "topographical maps, …, and satellite images (Table 1) in addition to various field observations and long-term temperature measurements." To make the distinction that Table 1 is related to these 3 products even more apparent.
New P4 L6.
Thank you, good suggestion. Implemented as suggested.

P5 L6 – First time DTM abbreviation is used it should be spelled out fully.
Done.

P6 L15 – "This resulted in a glacier area information for each data" doesn't make sense. Please clarify. Also, is this different than the first line of this section stating that glacier area was measured since 1859? It appears to be repetitive.
New P6 L1.
This might be a misunderstanding: it said "…in a glacier area information for each date" (not datA). We've adjusted the sentence to "This time series of maps and orthophotos resulted in a glacier area value for each corresponding date since 1859" to make it clearer.

P6 L24 – until 1997, and an additional data point…
New P6 L10.
The clause has been removed, because the GLAMOS measurements for Zmuttgletscher are not anymore used due to their problems of data gaps and reference positions before 1946.

P6 L32-33 – A bit unclear: were the two historic maps with debris cover symbols manually digitized as well? If so consider, "Debris cover extent for the orthophotos and historic maps that contained a debris cover symbol were manually digitized (Figure 2a and b)."
New P6 L19.
Yes, all were georeferenced and digitised. Changed as follows: "Debris cover extent for the orthophotos and historic maps was manually digitized." The fact that they contained the debris cover symbol is already mentioned before and repetitive.

P6 L33 – … Siegfriend map (1879) was verified using two photographs…
New P6 L21.
Done.

P6 L34 – This information was valuable …, which was the region of the strongest changes. Switching from past tense in first sentence to present tense in this last sentence. Suggest keeping the tenses the same for the paragraph to make it easier to read.
New P6 L22.
True. We've changed the second sentence to past tense as well.

P7 L5 – A bit confusing. First, you should reference Figure 1, since they show where these were provided. Second, what do you mean by "for setting the elevation change observations into context"? Do you mean you were doing this to compare to the elevation change estimates from aerial/satellite imagery? Or were you measuring the surface mass balance to be able to break the elevation change in the aerial/satellite imagery into the surface mass balance and the flux divergence? I assume this will become clearer in the results, but it should be clarified here as well.
New P7 L1.
It was done to have an idea of the melt, and to complement the elevation changes from DTMs with shorter-term field measurements.

We've changed the sentence to "Ablation was measured at seven points on the glacier tongue (**Error! Reference source not found.**) during summer 2017 to better understand the influence of debris on melt and to complement the longer-term information on elevation changes."

P7 L12-14 – A bit unclear as well. Do you mean "Debris thickness was measured via manual excavation along several transects perpendicular to the glacier tongue."? This seems to be the case from the figure and results.
New P7 L8.
Exactly. We've changed the sentence and added additional information: "Debris thickness data were collected in the field by manual excavation along and in between three transects perpendicular to the glacier flow direction in September 2017 (…)."

P7 L15-16 – "used to put observations into context" does not provide any information to the reader. What observations were you trying to put into context? What context? Additionally, stations had to be close to what? To each other? To the ablation stakes? Please clarify.
New P7 L13.
So many, so justified questions ☺ We hope it is clearer now. We've also added additional information that had before only been presented in the discussion.
New sentence: "The selected climate stations had to be as close to the study site as possible, lie at similar elevations, and cover a long period. Since no single station fulfilled all requirements, we used the stations in Sion (484 m, ~35 km distant) and Col du Grand St. Bernard (2472 m, ~40 km distant)."

P8 L28 – *which* reaches up to …
New P8 L13.
Done.

P8 L40-41 – consider consolidating the two sentences "Independent of the method, 100 m elevations bins (starting from 2100 m) were used to get a representative value for elevation change calculations in order to reduce the susceptibility of elevation change calculations to outliers due to the incomplete coverage of some areas of the glacier."
New P8 L20.
Thank you for the suggestion!
We've actually rearranged the whole paragraph since the explanation of the processing was maybe not quite clear:
"In areas of data voids or artefacts in the DTM, especially in higher elevations, no surface elevation change values could be calculated. In order to reduce the sensitivity to data voids and outliers, 100 m elevation bins (starting from 2100 m) were used for the elevation change calculation. In elevation bins without DTM coverage, the average of three extrapolation methods was used as the final mass balance value. The following three methods were used to fill the data voids: (1) a linear relationship between elevation and elevation change (ref), which is based on the strong respective correlation up to $R^2=0.93$; (2) using the same relationship but additionally setting the elevation change values to 0 when they become positive in the highest elevations; (3) using the mean elevation change value of the uppermost elevations that contain data and applying it to the glacierised area above."

P9 L22 – Do you mean, "change in debris thickness may impact the changes in surface elevation"?
As part of the consolidation process of the manuscript, this analysis was removed completely. The analysis was based on these measurements and the DTM difference maps of several periods. From the orthophotos and the elevation change maps it is pretty clear where thin and thick debris are located (very similar pattern today and in the past), as well as bare ice. But the uncertainty of the thickness classes for the earlier periods is large and therefore we decided to remove this analysis.

The message of this analysis is also contained in the ablation stake measurements, i.e. debris melt reduction.

P9 L25 – 20,000 m2? Or 20.0 m2?
Removed (see comment above).

P9 L37 – delete "additionally"
Done.

P10 L2 – The interferograms were…
New P7 L29.
Done.

P10 L4 – negative velocities were considered to be noise…
New P7 L32.
Done.

P10 L18 – this study.
Thank you for spotting. Done.

P10 L11 – The wording is a little awkward. Consider "From close to the end of the LIA (~1850s) to 2017, Zmuttgletscher has retreated by 1907 +/-12 m (12.1 +/- 0.09 m/yr). The maximum rate of retreat peaked between 1961 and 1977 at 21.7 +/- 0.04 m/yr."
Figure 3 – is there a reason Figure 3b, which is a table, is listed as a figure? What is the cause of the difference between Zmuttgletscher (this study) and GLAMOS 2018? They appear to be substantially different, albeit showing similar trends.
New P10 L2.
Thank you, we changed the sentences as suggested.
Former Figure 3 (new: Fig. 4) has been split up: the length change was slightly adapted (to distinguish the curves more easily) and kept in the main text as a separate figure. The two tables were moved to the supplement, where they still provide some useful information.
The difference between the two ZG curves cannot be well explained (as stated in the discussion 5.1). We agree that it is confusing and not necessary, since we have the length change information from our own data. Therefore we decided to remove the GLAMOS length record for Zmuttgletscher, as stated in P6 L10.

P10 L25 – "was with" does not make sense. "Until 2013" also sounds awkward. Consider something along the lines of "At the end of the LIA 2.8 +/- 0.2km2 of Zmuttgletscher was debris-covered, which has increased to > 5 km2 in 2013. During this time, the total glacier area has …"
New P10 L14.
Thank you. We've changed the sentence accordingly: "At the end of the LIA 2.8±0.2 km² of Zmuttgletscher were debris-covered (~12.9% of the entire glacier area), which increased to 5.03±0.1 km² in 2013."

P11 L4 – "all parts of the glacier" is fairly vague, perhaps in "all tributaries"?
New P10 L19.
We've changed the sentence to "Generally, the extent of debris has expanded up-glacier in TMG and SBG."

P11 L5 – the extent of debris has expanded …
Done.

P11 L5 – you refer to the debris-covered extent getting closer to Dent d'Herens, but isn't SBG (which is included in since you state "both" at the beginning of the sentence) getting equally as close to the headwalls of Dent Blanche?
New P10 L20.
Well observed, that was missing. We've added it to the sentence: "In both areas the debris extent has expanded to above the ice fall into the former accumulation areas and starts now close to the contributing rock walls of Dent d'Hérens (TMG) and Dent Blanche (SBG)."

P11 L4-10 – I think it's important to be specific when referring to properties of the debris cover. In this paragraph you are discussing the extent of debris cover, so this should be made clear. The "debris cover grew strongly" could refer to the thickness of debris cover or the extent of debris cover. Since both are discussed in this paper, it's important to always distinguish between the two.
Yes, this is important to avoid confusion. We have revisited all respective references and corrected them accordingly. Thank you!

P11 L8-10 – This sentence is very confusing. What does "after the exposure of the rock wall" mean? Please clarify.
New P10 L23.
We've changed the sentence and hope that it's clear now: "By 2010, the ice fall at STG had thinned sufficiently to disconnect and expose the rock wall beneath, from which a small rock fall detached between 2010 and 2013 (Fig 6). This debris mound covered an area of approx. 170x300 m and is now slowly transported down-glacier and spreading laterally"

P11 L18-20: difficult to read. Consider "In such cases, the typical base layer consists of fine-grained material (sand and even silt) lying directly on the ice, which is overlain by a few centimeters of pebbles. Above this typical base layer, there is no specific sorting."
New P11 L5.
Thank you for this suggestion. We have changed the sentences to: "Field investigations reveal a homogeneous debris cover in some regions with stone sizes in the top-most layer mainly between 10 and 30 cm in diameter, and a much more heterogeneous debris cover in other regions, with pebble-sized stones to boulders in the metre scale**Error! Reference source not found.**. The typical base layer of the debris consists of fine-grained material (sand and even silt), and is overlain by a few centimetres of pebbles."

P11 L12 – You state these are over several periods, but the caption states these measurements are from 05/07 – 22/08 2017, which is a single period in time. Please clarify.
New P11 L16.
True. We have corrected the text accordingly: "Ablation measurements from seven locations and over a seven weeks period in summer 2017…".

P12 L20 – what does "number" refer to? The number of ice cliffs?
New P12 L20.
We've changed the sentence to: "Even though the glacier has become more and more debris-covered, we did not find any clear trend in area and location of ice cliffs (Table 3)."

P12 L21 – "lower parts of the glacier tongue until the terminus" is redundant.
New P12 L21.
We've changed the sentence to: "They are almost exclusively located in the lowest part of the glacier tongue (yellow area in Fig9)."

P13 L2 – delete "only very"
Done.

P13 L10 – delete "only"
Done.

Figure 8 – there appears to be a line along the top of the figure?
Maybe it's a display error. I've exported the figure again, it should be fine now.

P13 L19-20 – A bit redundant saying in some periods, then stating the period, and then doing the same later. Additionally, I'm not sure "strong" is the best choice of words to describe elevation change. It'd be better to be specific: "most negative" or the "surface lowering was greatest", something along those lines. For example, "The elevation change was most negative near the terminus between 1946-1977. Since 2001 though, the elevation change over the tongue has become more heterogeneous."
New P13 L13-25.
We have rewritten the whole paragraph and included your suggestions on terminology. We hope it is now more concise, easier to read, and less repetitive.

P13 L22 – "Between 2879 and 1946, the *average* surface elevation change was -0.63 m/yr and most pronounced at the terminus. *Surface lowering increased* to …". Note: This is a bit repetitive of L19-20.
See comment above. Repetitive content was removed.

P13 L26 – the tongue's *average* surface elevation *change* was almost balanced at -0.11 m/yr.
New P13 L17.
Sentence changed to: "Between 1977 and 1983 the tongue's average surface elevation was almost stable at change rates of -0.11 ±0.29m yr-1."

P14 L1 – use either "surface lowering" or "surface elevation change". Previously, you've used "surface elevation change", so be consistent. This is especially true because you're using positive and negative values for elevation change, so surface lowering with a positive value makes this difficult to read.
See comment above. Paragraph rewritten and suggestions implemented.

P14 L3 – *slight*
New P13 L19.
Done.

P16 L4 – do you mean the ratio of thinning for thick debris compared to bare ice?
As part of the consolidation process of the manuscript, this analysis was removed completely (see comment above).

P16 L14 – please clarify what "no clear hint of an increasing balancing effect" means? There is no clear trend in the mass balance gradient flattening towards the terminus?
remove the statement? Because further down we say there is an effect of the debris cover…
New P13 L26.
We have re-written most of the subchapter, among others removed this statement, which had even been somewhat contradicted by the text below.
Additionally, we agreed with R2 that when we want to assess the elevation change gradient it makes more sense to look at the changes of elevation-based classes than of spatial sections. Therefore, we have changed the analysis to look at elevation changes of 50 m classes. Additionally, we have reduced the number of periods from eleven to seven. Thus, the additional map as well as the table are not necessary anymore. The whole analysis should now be easier to understand and the figure easier to read.

P16 L24 – again, this appears to be a ratio. Ratios are not very intuitive. For example, if both regions were positive, they could still have the same ratio. Why are you reporting and discussing the ratio and changes in ratios? Is it providing information about how "connected" the upper and lower portions of the glaciers are? Is it used to understanding the dynamical state that the glacier is in? Please add a sentence or two to clarify what the ratio is actually telling us about the glacier. See comment above.

P16 L26 – *still show*
Sentences changed (see comment above).

Section 4.5 – Is this referring to the "glacier-wide" mass balance? If so change the heading. New P9 L5.
It refers to glacier-wide mass balance indeed. We think this should be inherent in the word 'geodetic', as it needs to be done glacier-wide to be able to calculate the mass balance. We made it clear that it is done glacier-wide in the respective methods chapter.

P17 L1 – showed an *increase in elevation* …
New P9 L9.
Done.

P17 L2 – After 1988, the mass balance became more negative again, even while the lowest areas on the tongue had a stable or positive mass balance.
New P9 L10.
Changed.

P17 L4 – When did this negative trend intensify? Rearrange this sentence if it was in 2001.
New P9 L11.
It intensified after 1988. We've changed the sentence to "The negative trend continuously intensified and after 2001 …".

P18 L4 – add the year that this was done again for reference, so that it is clear in the text. It is obvious this this is an independent measurement, so you can delete "as an independent measurement"
New P12 L14.
Done.

P19 L13 – and *thus have* little impact …
New P15 L9.
Done.

P19 L14 – delete "also"
New P15 L10.
Done.

P19 L31-L36 – this reads as a methods section, i.e., the reason you chose the methods you did as opposed to a discussion section. Consider moving this to the methods, although I leave this choice up to you. The other ones "suitability" of methods seem to discuss other studies, shortcomings, etc., which is why I don't think the other parts of this subsection are out of place.
New P15 L25.
We have moved parts of this paragraph up to the methods section (P6 L10) and also shortened it, but kept the part about uncertainties in place. We hope that the division is clearer now.

P19 L39 – "Nevertheless we think…" are you referring to the long-term trend or are you saying that despite potentially underestimating the area, you still think the trend is significant?
New P15 L25.
Yes exactly. We have rephrased it to: "For 1961, the debris cover extent might be slightly underestimated as the orthophoto does not fully cover the ablation area and a thin snow-cover was present in higher areas. This could partly explain the slight reduction in debris extent from 1946 (3.98±0.01 km²) to 1961 (3.79±0.01 km²)."

P20 L13-5 – "similar behavior", perhaps similar trend? "pointing out" also sounds awkward here. Perhaps "Zmuttgletscher has shown a similar trend indicating that its glacier-wide mass balance has foremost been governed by climatic changes rather than a response to changes in debris cover."
New P18 L27
We've changed the whole paragraph about the climatic evolution and incorporated this statement in a different way. The closest sentence is: "The evolution of glacier-wide mass balance of Zmuttgletscher since 1879 is on century and decadal time-scales comparable to trends on other debris-free glaciers in the Swiss Alps, which are closely related to variations in climate (Fig 3, refs)."

P20 L31 – Be careful with "higher" elevation change rate as this could imply a positive elevation change, but here it seems to be referring to being more negative? Please clarify.
We have changed the terminology in the whole paragraph. It should be clear now.

P21 L3-5 – A bit confused here as to how the debris-covered area was stable if the upper margin of the zone was moved down-glaciers? Are you suggesting that the glacier advanced and that's what kept the debris-covered area constant? Otherwise, if the upper margin, which I assume is referring to the debris/ice extent interface moved downglacier, how could the debris extent remain constant?
New P16 L19
Yes, the way the sentence had been phrased was confusing. What happened was that only in the central part of the tongue the clean-ice areas were pushed down and widened. We have revised the sentence to make this clear: "At the same time, the upper boundary of debris cover in these areas migrated slightly down-glacier (Fig. 12a), but because the debris cover continued to expand in other areas, the total debris-covered area was roughly stable during this period."

P21 L8-10 – "again" typically does not follow the verb, e.g., became more negative again, or the debris-covered area strongly increased again.
Thank you. We have adjusted the word order in several places throughout the document.

P21 L23 – and *also likely* thicker debris
New P19 L31
Done.

P25 L18 – delete a period
Done.

P25 L18-21 – In Section 5.2.1 you state that the magnitude of mass change is governed by changes in climate as opposed to changes in debris cover. However, here you are stating that the different magnitudes in mass balance between Zmuttgletscher compared to glaciers in the Himalaya is likely attributed to the differences in debris cover. While this could play a role in the differences between the two, how do you know that this difference is not caused by differences in climate forcing?

Correct, it could also be the case for the Himalayan glaciers that debris governs the pattern rather than the magnitude of mass changes.

The comparison to glaciers in the Himalayas has been cut down considerably. Among others, this statement was removed.

P26 L6-14 – Can this be consolidated? While it is nice to place the changes in debris-covered area into a regional context, the point gets a bit lost in this long list of every study, region, and change.

We have now strongly cut down this section and at the same time included more information about the absolute debris cover from other studies. At the same time we think that the point has become clearer. Most references have been kept for interested readers.

P27 L5 – "re-thickening"? It thickened, so why include the "re-"?

New P21 L16

True, it doesn't provide any additional information. We've removed it.

P27 L7 – These observations prove a …

New P21 L20

Changed to "These findings suggest a clear and direct …"

**Response to reviewer comments**

Reviewer #2

**Specific comments**

P1L1
The title is a tad misleading, maybe. Yes, debris cover is incorporated in the analysis and a major component, but the main message is the evolution of the glacier. The results even show that the overall mass evolution of the glacier is not strongly affected by debris? I'm thinking something in the line of "Post-LIA evolution of Zmuttgletscher and its debris cover".
Fair remark. We changed the title to
'Unravelling the evolution of Zmuttgletscher and its debris cover since the end of the Little Ice Age'

P1L6
Often -> generally
Flat. Debris-covered glaciers are generally not flat. They are hummocky due to the spatially variable melt rates induced by the debris. I think you mean gently-sloped or of low gradient.
New P1 L8
True. We've changed 'flat' to 'gently-sloping'.

P1L7
Today -> at present
New P1 L8
Changed to: 'At present, many of these glaciers show high thinning rates despite thick debris cover.'

P1L11
Increased from approx. 13% to more than 32%? Provide specific numbers.
New P1 L13
We are of the opinion that the abstract should provide a rough picture of the results and at the same time be as concise as possible (also required by the journal). Therefore we kept the rough numbers as exact numbers would also need the uncertainty range, which makes the sentence longer and less easy to read.

P1L15
Maybe provide numbers for the area and cliff changes.

~2005; ~1.5 -> Again be specific with you numbers.
New P1 L19.
Thank you. We've removed the tilde as these numbers are as precise as they can be.

P1L20
Why not just call it introduction?
New P1 L25
Isn't this just a question of personal preference? I just think 'motivation and objectives' is a bit more explicit. In any case it takes up one line.

P1L25
Similar rates of thinning at the same elevation bands. Volume changes would imply mass balance but these are largely unknown from these studies. Also no reference to the (more recent) elevation difference paper here [*Brun et al.*, 2017]?
New P1 L30
Changed to "… exhibit similar thinning rates as debris-free glaciers…".

As far as we can see, Fanny Brun et al. 2017 did not distinguish between debris-free and debris-covered glaciers and didn't make any respective statements. Therefore it would not be correct to cite this paper in this context.

P1L26-29
There is some debate the last years about the importance of supraglacial ponds, ice cliffs and glacier dynamics/emergence. I think there are a number of recent papers that could be added here that touch upon this topic, e.g. [*Pellicciotti et al.*, 2015; *Vincent et al.*, 2016; *Brun et al.*, 2018; *Miles et al.*, 2018].
New P1 L35
Correct, we've added a few of these. There would be even more interesting studies, so we've also added an 'e.g.' before the references.

P1L37
Wouldn't call this 'as a result'.
These glaciers also often have a long flat (low-gradient!) tongue because the insulating surface debris layer just allows them to extend into the lower, warmer valleys.
New P1 L37
We have re-written the sentences so that it becomes clear that all these things are linked. The fact about the lower elevation of the glacier tongue is also linked to the reduced ablation, which was mentioned two sentences before we tried to summarise the above in the sentence that starts with 'as a result'. We hope this is clear now.
We have changed it to: "Especially during periods of negative mass balance, the down-glacier increase in debris-cover thickness reduces ablation through its insulating effect and can lead to a lower – and even reversed – mass balance gradient (refs). Over time, this reduction in ablation can lead to a decrease in surface slope and, consequently, driving stress and ice flow velocity (refs). Furthermore, with an increase in equilibrium line altitude (ELA) the englacial debris melts out earlier, leading to an extended debris cover further up-glacier (refs). As a consequence of reduced ablation and driving stress, heavily debris-covered glaciers often have long, gently-sloped, and low-lying tongues with low flow velocities or even stagnant parts."

P2L1-6
Long. Rephrase.
New P2 L6-11
We have rephrased the sentence to: "Current research at many debris-covered glaciers and specifically in the Himalayas is mostly focussed on processes, such as ablation beneath the debris cover and in areas of ice cliffs and ponds, thinning of glacier tongues, surface flow velocities, or changes in debris cover over time (refs). A few studies have started to integrate the existing understanding and interactions of some of these processes into numerical models for the evolution of such glaciers (refs)."

P2L7-12
Same here. Basically two paragraphs of one sentence each. Also these long itemizations could use some inline numbering.
New P2 L12
Very good idea, we have added a numbering of the different shortcomings to distinguish between the main items of this paragraph:
"However, most studies … of debris-covered glaciers: (i) time series are often short …; (ii) investigations are often local …; (iii) by considering only one or few variables … ; (iv) studies at glacier scale … ."
However, grammatically, the paragraph consists of five 'sentences', not one. A semicolon can be used to separate independent clauses, the content of which is closely related. We consider this an element of style and less important for the understanding of the paragraph and therefore prefer to keep it.

P2L22
I think the importance part can be skipped, not quite relevant here and statement has an endless list of footnotes.
Yes, true. We've removed it (" – which are in the center of attention due to their importance – ").

P2L36

Are the latlon for the peaks relevant? The only important one, for the glacier, is missing.

New P3 L2-5

The one for the glacier is in Figure 1a, but we've added it to the text. It's also true that the important information is rather the elevation of the peaks than their coordinates. We've removed them, making the text also a little bit nicer to read.

P2L40

Only originates from the rock walls? What about the other possible sources, see e.g. [*Evatt et al.*, 2015]?

P3 L5

You're absolutely right. In the discussion we also state that there are potential other sources (like moraines). We've added 'mainly': "The debris mainly originates from the surrounding rock walls." However, we've not added Evatt et al. (2015) because it's rather a theoretical consideration about the energy balance within the debris layer than actual analysis of debris sources. Therefore, in the discussion we stick to van Woerkom et al. (2018).

P3L9-10

"at near-by almost"???

Can similar values really be assumed. We often considerably different behaviour of glaciers in a valley due to differences in microtopography etc.

New P3 L8

We have changed the sentence to: "There are no direct measurements at higher elevations, but model estimates suggest values between 800-1500 mm (ref). Glaciological mass balance measurements at the nearby debris-free Findelgletscher (15 km distant) suggest end-of-winter accumulation around 0.8-1.5 m water equivalent (w.e.) (ref). However, at Zmuttgletscher, avalanching additionally contributes to accumulation."

The measurements from Findelgletscher are the best guess we have at our disposition and we think it is valid to provide this information. However, we have rephrased the sentences to make clear that this is just a best guess and not the exact numbers we assume for Zmuttgletscher.

P3

I think that there is some irrelevant information provided in the study area section. Keep it clean and simple.

We have taken out information about climate, geology, and other literature. Thus, we could shorten the chapter from 662 words to 452 words.

P4L9

plane -> airplane

New P4 L6.

Done.

P5 section 3.1

Because the results depend strongly on the DTMs I think this section is a bit short. There are all kinds of caveats and things that can go wrong with DTM generation (especially working without fiducials and markings in Agisoft). A bit more detail on the exact procedures followed would be welcome.

New P5, section 3.1

We have provided some further details. However, the generation of the DTMs is described in detail in a paper by Mölg and Bolch (2017), including a quality comparison of the output of two SfM software packages and a photogrammetry software (ERDAS Imagine). Among others, the idea of that paper was to have a base to refer to in order to save spae in the methods description in the current study.

P5L25

The DTM was produced FROM the tri-stereo image using photogrammetry, I'd guess.

New P5 L21
True, here the egg laid a hen. We've changed the sentence to: "In addition, we generated a glacier-wide DTM for 2017 (spatial resolution 1 m) using Pléiades tri-stereo, high-resolution satellite imagery based on photogrammetric principles using PCI Geomatica (ref)."

P6L8
manually digitizing / manual digitization
New P5 L29
Done.

P6L15
I think calling this ice fall is quite confusing terminology. Maybe ice deposits?
New P6 L5
We've changed the sentence to: "…since they contribute mass to the main glacier through frequent ice avalanches."

P6L17
…allowed correct interpretation of…
New P6 L3
Done.

P6L29-30
Isn't this just normal error propagation? If it is not, why did you choose to
New P6 L16.
Yes, indeed. Isn't this what we say? The sentence is: "For the calculation of the changes, the uncertainties of the two respective dates are combined by error propagation, analogue to the mass balance data (ref).

P6L32
which -> that
New P6 L19
We've changed the sentence to: "Debris cover extent for the orthophotos and historic maps was manually digitized (Fig 2)." We've removed the 'debris-cover symbol' since it is already stated before.

P7L5
Undertook -> performed
New P7 L1
We've changed the sentence to: "Ablation was measured at seven points on the glacier tongue during summer 2017 to better understand the influence of debris on melt and to complement the longer-term information on elevation changes."
for setting -> to put
see above

P7L6
…two metre long PVF stakes…
New P7 L3-4
We've changed the word order.
Why 'PVF'? I think the abbreviation 'PVC' comes from 'polyvinyl chloride'. But maybe this was just a typing mistake.

P7L12
information -> data
P7 L8
Done.

P7L12-14
When and how often were these measurements performed. Need more details.
New P7 L8-11
The data was taken during a campaign in September 2017. We have changed the paragraph as follows and hope it is clear now: "Debris thickness data were collected in the field by manual excavation along and in between three transects perpendicular to the glacier flow direction in September 2017 (for transect locations see **Error! Reference source not found.**; results of upper transect see **Error! Reference source not found.**a). In general, each data point represents the average of three measurements ~1 m apart, and the standard deviation is used as uncertainty measure. For debris thicknesses above 20 cm only one measurement was taken."

P8L1-6
Have you not considered the object-based ice cliff mapping I've done before [*Kraaijenbrink et al.*, 2016]?
New P7 L39
Yes, we have. We have now included a reference to your paper in the methods section.
The problem of a more sophisticated approach, as you described it, is that we had very variable input data, mostly black & white, with different texture, quality, and resolution. Also, the link to topography was very weak, i.e. there are ice cliffs with a relatively small slope (<25°) and steep areas (>40°) that are still debris-covered. The most important issues are discussed in the methods discussion chapter.

P8L17-18
I do not understand what you mean here. How can one assume a plane due to curvature? Or is the plane curved to mimic the laterally slightly convex glacier surface? But then it is not a plane, right?
During the consolidation process of the manuscript this section has been removed because it had not been used for further analysis.

P8L32
I'm not sure if "stand" is the right word here
Included in the removed sections, see comment above.

P8L36-40
It is completely unclear to me when each specific method was used and for what reasons.
New P8 L19-29
It is simpler than it sounded: all methods were used for each mass balance period. For elevation bins with no coverage, the three extrapolation methods were applied and the average of the three methods was taken as final value, while the standard deviation went into the uncertainty calculation.
We've changed the structure of the whole paragraph and hope that it's clear now:
"Glacier-wide geodetic mass balance was calculated between dates for which DTMs covered large parts of the glacier surface (1879, 1946, 1977, 1988, 2001, 2010, 2017). In areas of data voids or artefacts in the DTM, especially in higher elevations, no surface elevation change values could be calculated. In order to reduce the sensitivity to data voids and outliers, 100 m elevation bins (starting from 2100 m) were used for the elevation change calculation. In elevation bins without DTM coverage, the average of three extrapolation methods was used as the final mass balance value. The following three methods were used to fill the data voids: (1) a linear relationship between elevation and elevation change (refs), which is based on the strong respective correlation up to $R^2$=0.93; (2) using the same relationship but additionally setting the elevation change values to 0 when they become positive in the highest elevations; (3) using the mean elevation change value of the uppermost elevations that contain data and applying it to the glacierised area above. The area-weighted average elevation change of all bins was summed up to reach the average elevation change of the total glacier surface."

P9L2-3
Just a complicated way to say area-weighted?
New P8 L27

Correct. We've changed it to: "The area-weighted average elevation change of all bins was summed up to reach the average elevation change of the total glacier surface."

P9L10-11

I do not understand why this was done. Now you're assuming a single elevation for the entire period for specific parts of the glacier? Wouldn't it be better to leave it as no data and perform weighted statistics appropriately?

We have removed this section since it adds complication and confuses and is irrelevant for the total uncertainty.

For the record and in case you are interested: We did not use the elevation of 2010 for the entire period, which would indeed introduce a relatively big error. Instead, we only used the AREA of the elevation bins from 2010 and applied them to those dates, where the DTMs did not cover all elevation bins.

P9L22

impact on surface -> impact surface

This analysis has been removed in the revised manuscript.

P9L23

So there is a class for 5 cm debris and for debris thicker than 15 cm. What about the rest? That is, between 5 and 15? Or do you mean just two classes, thin and thick, <15 and >15?

As part of the consolidation process of the manuscript, this analysis was removed completely (see comment below).

P24-P26

So now there is suddenly a lot of debris thickness data. I don't understand this? How were the maps produced? Were there that many pits dug for all these time steps? If so, that's quite impressive.

The only debris thickness measurements are from 2017. The analysis was based on these measurements and the DTM difference maps of several periods. From the orthophotos and the elevation change maps it is pretty clear where thin and thick debris are located (very similar pattern today and in the past), as well as bare ice. But the uncertainty of the thickness classes for the earlier periods is relatively large and therefore we decided to remove this analysis. The message of this analysis is also contained in the ablation stake measurements, i.e. debris melt reduction.

P9L35

Correlation quality ('strength')? I'd just use 'correlation'. Correlation itself implies a quality/strength of fit.

Yes, absolutely. The term 'Strength' referred to the output parameter of this name by the software. It is confusing, thus we have removed it.

P10L8

Why *smaller or equal* than 0.03, and not just a uncertainty value?

New P7 L36

Changed to ±0.03 m/yr.

P10L8

There should be no space between the first number and the plusminus symbol.

You use m/yr semi-consistently throughout the manuscript. Preferebly use scientific notation and in glaciology it is somehow common to use pro annum instead of per year: 12.1 m/yr -> 12.1 m a-1. Change this throughout.

We've removed the space throughout the manuscript.

We've changed m/yr to m yr-1 (as it is used by many others, e.g. Benn et al. (2012), Fischer et al. (2015), Brun et al. (2017) etc.). This way the abbreviated words are consistently in English (which I find somehow more intuitive to read).

P10L14

slowed down -> decelerated
New P10 L5
Done.

P10 Fig3
Figure 3 is a bit confusing with all the length changes and additional table. What is the point here. Maybe combine figure 3 and 4, and skip the display of the other glaciers? Just mentioning in text that they are quite similar around Switzerland would suffice, I think.
New P10 L8
It is true that the combination with the tables was a bit confusing. We have removed the tables and shifted them to the supplements, where they still provide useful information (esp. on area changes). Nevertheless, we have kept the length change figure. It was slightly adapted to be easier to distinguish the curves. This length change curves of the different glaciers – which are actually quite different – are imortant for the interpretation of a debris-cover effect in the length change. Among others, this figure is used in the discussion to arrive at this conclusion.

P10L27
…resulting in 33% debris cover at present.
New P10 L15
Sentence changed to "During this time the total glacier area has decreased from 21.24±0.4 km² to 15.82±0.3 km², resulting in 31.8±2% of the glacier area being debris-covered in 2013 (fig)."

P11 Fig5
The different periods are a bit difficult to read with the current colour scheme. This could be improved by mapping it over a wider range of luminosities, i.e. from lighter yellows to darker reds.
New P11 L1
We agree that the colour palette was not ideal. We've changed it to run from dark brown to orange to yellow.

P11L16-17
I'm not a geomorphologist, but is superficial the right terminology?
"in the metre scale" sounds strange
New P11 L5-7
Yes. We've changed the sentence to: 'Field investigations reveal a homogeneous debris cover in some regions with stone sizes in the top-most layer mainly between 10 and 30 cm in diameter, and a much more heterogeneous debris cover in other regions, with pebble-sized stones to boulders in the metre scale …'

P12 Fig7 panel b
No error bars on these points?
The melt rate below the debris surface depends both on elevation (i.e. temperature + radiation) and on debris thickness. To have an accurate curve the melt rates have to be normalized by the clean ice melt rate at the same elevation. Does *normalized* in this figure mean that you had a stake at the same elevation on a clean part of ice, representatively close to the debris stake, that was used to normalize each point? If so, did you clear the glacier of debris or did you find a naturally clear spot? Could there then have been errors in the clean ice ablation measurements by radiation emitted or reflected by nearby debris; an energy balance component that would not exist on a completely debris-free glacier?
EDIT: I see now in the text there was one reference stake at 2600. I'm then not quite sure whether the conclusion made are sound.
New P11 L11
Correct, the reference stake used for the normalisation is the debris-free stake on 2600m. This is the lowest elevation with clean ice. Our conclusions are that (i) debris of the observed thickness does substantially reduce ablation, and that (ii) the ablation rather depends on debris thickness than on elevation. Looking at the individual stakes we see zero correlation between elevation and melt, but an R2 of 0.95 when we apply a logarithmic function to the measurements above 0 cm thickness. Also, the

measurements for stakes with 5 and 25 cm yield almost the same melt rate even though the stakes are in quite different elevations. All measurements were taken on roughly flat surfaces. Since we do not use the derived Östrem-curve for further analysis, the exact numbers do not have any impact. But we think there is sufficient evidence to arrive at the conclusions mentioned above and in the text.

We have changed to figure to show melt rates over the 7-weeks period. Thus, we arrive at the same conclusion without introducing a small additional error by normalising with the clean-ice stake from a different elevation.

We also added a sentence about the uncertainties. They are too small to display as error bars in the figure.

P12L16-17
Could perturbation of the debris layer during drilling of the stakes have caused a difference in the debris matrix that could have affected the melt rates?

We gave our best to replace the debris the same way it was before, but some deviation from the previous sorting cannot be excluded.

We do not know how this difference could be assessed. Do you have an idea? But if there was a substantial difference, we would have seen a deviation of the melt around the stake and the immediately surrounding area, which was not the case.

P12 section 4.3
I am a bit confused by the mixed terminology between exposed ice and ice cliffs. At first I thought that with exposed ice, bedsides ice cliffs also patches with very thin or absent debris were meant. However, in this section and also further in the manuscript, it seems like exposed ice and ice cliffs are used interchangeably. Please make this clearer and be consistent with your terminology. If you just mean ice cliffs, I would suggest sticking to that term, since this is most commonly used in literature.

We've changed the terminology. We made clear in the methods that the term 'ice cliffs' includes both exposed ice from ice cliffs and exposed ice from water flow channels and then stuck to the term 'ice cliffs' throughout the document.

P12L23
,but the -> ,and

This statement has been removed.

P13L1
The promille is a bit confusing and unnecessary here, I think. Just stick to 0.5% and 1.8% in this sentence.

New P12 L23

Yes, confusing indeed. Changed as suggested.

P13L10
This also depend on variable rates of ice emergence, which should theoretically be taken into account if an analysis on a subset of a geodetic dataset is performed, also see [*Brun et al.*, 2018]. Ideally, to really look at the effect of ice cliffs and their relative melt, there should be some correction for the downglacier displacement of the ice and cliff.

New P12, section 4.5

The objective of this analysis is to understand the potential influence of ice cliffs on elevation change. To do so, we compared area-averaged elevation change with and without ice-cliff and their surrounding areas but could not see substantial differences. Importantly, we do not explicitly compare different time periods. Also, both horizontal displacements and cliff backwasting are inherently accounted for by applying the buffer. Thus, our main conclusions only refer to elevation change per date (not ablation) and are in our view robust and do not require emergence velocities and horizontal displacements.

We adjusted the description to make the objective of the analysis and the conclusion drawn from it clear.

P13L26
balances -> stable

New P13 L17
Sentence changed to: "… the tongue's elevation barely changed…".

P13 section 4.4
Uncertainty ranges should be included in the numbers that are provided in this section.
New section 4.6
Done.

P14L7
Pushed down sounds odd. "Travelled downglacier with the flowing ice"?
…at a higher rate…, is it relevant if the rate is higher? They are 'pushed' irrespective of the velocity (expect for completely stable ice).
We have removed this statement since it doesn't provide any crucial information.

P15 Fig9
Color of the class -0.5–0.5 should be white, in my opinion.
New Fig 10, P14
The problem with white is that it cannot be distinguished anymore from the background, i.e. where there is no data available. We have changed the yellow into yellow-grey to give it a more neutral appearance. Too grey would again move it to the blue category, thus it's a fine line.

P16L6
"at 50 cm resolution"
This sub-chapter has been completely removed. The analysis of the difference between the two UAV DEMs was removed because it contained a redundant message.

P16L11
remove 'due to higher temperatures'
We've removed the sentence, since it was a repetition from above.

P16L14
"There is no clear hint" does not sound very scientific.
New P13 L26-28.
True. We have strongly changed the text and reduced it to the central content. Also, we incorporated it into the elevation change chapter (4.6), since it is an extension of the same data and also about elevation change patterns.

P16 Fig10
Instead of plotting glacier section number on the x axis it would be more informative to use the mean elevation of a section instead. In that case, you may also consider switching the axes to get an dh/dt gradient kind of plot, similar to figure 17.
What does the 'relation' show in the subtable? A regression through sections? Should ideally be filtered for significance. Consider skipping the table entirely.
New P15 L1
We agree that when we want to assess the elevation change gradient it makes more sense to look at the changes of elevation-based classes than of spatial sections. Therefore, we have changed the analysis to look at elevation changes of 50 m classes. Additionally, we have reduced the number of periods from eleven to seven. Thus, the additional map as well as the table are not necessary anymore. The whole analysis should now be easier to understand and the figure easier to read.

P16L29
This is not based on your data, right? Provide inline reference.
New P9 L5
Probably this is some kind of misunderstanding. Here we present the results of our mass balance analysis.

P18L4
Unclear, please rephrase.
New P12 L14
We've rephrased the sentence to: "The displacements from radar interferometry from the 1.5-day period in summer 2017 yield similar results and confirm the quasi-stagnation of the lower tongue …".

P18 section 5.1
Could use some subsubheadings
New P15 L6
We chose to keep it separated by the different paragraphs. Subheadings would take up additional space and the chapters would mostly be rather short. The chapter has also been shortened considerably (from 913 to 600 words).

P19L9
These uncertainties are a bit unclear to me. A range from *about* 1 meter to 2 meter. Also, when you convert that to m a$_{-1}$, doesn't that depend on the variable time span between each observation pair?
New P15 L7
Here we provided the range because the uncertainties are different for each DTM. We added a reference to Table 1 where these uncertainties are stated.
Yes, the value of the annual uncertainty depends on the length of the time span. We considered this accordingly whenever we provided annual change values.

P19L19-20
Bit irrelevant background info
Deleted.

P19L27-30
I agree. I think it is not fair to compare 1.5-day measurements to those over much longer time spans. There will be quite some intraannual, intraseasonal and probably diurnal variability in velocity. This should clearly be acknowledged prior to the discussion.
New P15 L23
We have kept this information at this location where we think it is appropriately placed. Also, the radar information is used as complementary data to get a preciser idea of the velocity pattern and the stagnating glacier part. For this purpose the hint in the discussion to the seasonally restricted meaning is sufficient.

P19L42
Don't miss [*Kraaijenbrink et al.*, 2016] here :-)
New P15 L30
How could we ☺
We agree that it was an important reference that we missed to cite.

P20L8
I don't think the discussed P and T encompass "*all* glacier-related variables".
During the reorganisation process, this statement got removed.

P20 Fig 14
Not a big fan of introducing new figures, data and analyses in the discussion section.
Discussion is to discuss the results already presented.
We agree. Newly, we included the temperature change information in the mass balance figure. By sharing the same x-axis, the two pieces of information can nicely be linked, even visually.

P20L22
is it a constant increase, or a continuous increase?
New P16 L7

We've restructured the sentence to: "At the same time, the debris-covered area increased from 2.8±0.2 km² in 1859 to 3.79±0.01 km² in 1961"

P20L28

Exposed-ice? You mean ice cliff? Earlier comment applied to this section as well.

We changed the terminology (see earlier comment). It is now 'ice cliff' in this case.

P21L4

Was even moded -> has even moved. Same next sentence.

New P16 L21

Done.

P21L7

Attenuated is not the right word here. Attenuate means *to make something else smaller, thinner, or weaker*.

P16 L23

True, thanks for spotting this. We agree that the term is not quite adequate in this case. We've changed it to ".. the retreat rate and area loss gradually slowed down."

P21L14

During…decades -> Over the last 16 years

I really don't get these ranges that use the tilde symbol. About 0.80 to 0.98 seems rather specific to me. Why is this an estimate?

New P16 L33

Changed to 'In the most recent period (2001 to 2017) …'

We tried to keep the discussion on a qualitative level as far as possible. Exact values here would mean the values of three periods including uncertainty ranges, which is long, repetitive, and not really necessary. Therefore we kept the rough values but exchanged the tilde symbol with an 'approx.'

P21L15

Same for larger than a range. Isn't >1-2 m the same as >2, essentially? Or do you mean something like greater-than above similar or equal to 2 m? (e.g. $\gtrsim, \gtrsim, \gtrsim$)?

New P16 L33

True. We changed it to >1m yr$^{-1}$

P21L16-17

"As a result length and area changes have been comparably small given the high mass loss".

New P16 L35

Thanks for the suggestion! Sentence changed to : "… which might explain the relatively small length and area changes given the high overall mass loss (Figs)."

P21L26

"surfacing of debris"

During the revision, this statement has been incorporated in other sentences.

P22L17

We always refer to these cavities as *voids*, as per i.a. [*Benn et al.*, 2012]

New P19 L41

We changed 'cavities' to voids as suggested.

P24L26

"A sample" is not very specific. Also why suddenly this new analysis, which does not really provide any novel insights, at the end of the discussion?

New P17, chapter 5.4

We have restructured the discussion such that the overview is clear and the sections are linked to respective main messages. This analysis clearly yields that debris-free and debris-covered glaciers can be separated based on elevation change patterns, and that Zmuttgletscher can clearly be grouped to the latter. We have re-structured the discussion to put this analysis in a better position to be used for argumentation.
The y-axis of the figure shows only elevation bins and no absolute numbers because the absolute elevation was normalised for comparison. This is necessary to account for the different elevation ranges of the glaciers. To make this clear we will change the axis caption from 'Elevation class' to 'Normalised elevation class'.
We also prefer to keep this figure in the discussion since this is the place where we conclude the effect of debris cover. Additionally, this is not a real 'analysis' per se: we have simply used existing DTM data (prepared by Fischer et al. 2015) and plotted it in a comparable way for a selection of glaciers. Thus it would not really fit into the 'results' chapter, and doing so would also lead to additional repetition, which we tried to avoid as much as possible in the revised manuscript.

P24L29
Remove "than higher up"
Done.

P25 Fig 17
I think you could do without this analysis and this figure. If you decide to keep it, at least indicate the actual elevation instead of class number.
See comment above. The data was normalised, so we can only adjust the axis title, not use actual elevations.

P25L9
Swiss glacier's mass balance -> Swiss glaciers
Sentence removed.

P25L15-17
per year is missing from the units here
Thank you! Attended.

P25L23
remove today
Done.

P25L32
contributary cause -> contributes to
Done.

P26L1-…
I found the sudden use of procent points a bit strange in this section. Points are a bit irrelevant as they do not show whether something changed from for example no debris to 30% debris, or from 60% to 90%. Surely the actual increases are presented in the original papers?
New P17 chapter 5.3
We chose the percentage points because it is easier to compare numbers and the reader doesn't need to calculate, because otherwise it would be necessary to present the debris cover share both at the start and the end of the period. We have now strongly cut down this section, incorporated it into a separate chapter about debris cover changes, and at the same time included the missing information about the absolute debris cover from other studies, which covers a broad range: "These studies found debris cover increases of 2-10 percentage points with a broad range of absolute debris cover from 2-42%."

In case you're interested in the absolute numbers, here are the debris cover changes from studies:

Kellerer-Pirklbauer 2008: 5.4-7.3%, 1964-2000
Stokes al 2007: different glaciers, Djankuat: 6-9%, others from 3-7%, 2-7%, 15-17%, 26-30%, 1985-2000
Bolch al 2008: 39-42%, 1962-2005
Bhambri al 2011: 21-26%, 1968-2006

[revised manuscript text omitted]
 heterogeneity is even increased by the presence of ice cliffs. overall Debris cover has decreased the
elevation change gradient over time to almost zero. The role of Ddebris-related surface features on
Zmuttgletscher**

Whilst the presence of a debris mantle may not substantially influence the overall mass loss rate of Zmuttgletscher, we
concludefind that the main reason for the observed heterogeneous thinning pattern is the increasingly extensive and also
likely thicker debris cover that is heterogeneously distributed over the tongue. The heterogeneity is evenfurther increased by
the presence of ice cliffs. Judging from field Field visits and patterns of surface lowering suggest a close association between
ice cliff formation and the presence of DEM differencing, the most relevant surface features leading to exposed ice areas
such as ice cliffs – and hence enhanced ablation – on Zmuttgletscher are supraglacial meltwater channels. Surface melt water
is often running off superficially over long substantial distances and several water flow channels are abundant exist all over
the glacier tongue, outside of crevassed areas. Inside and alongside these channels bare ice often becomes exposed when
water washes away the debris or laterally cuts into the ice (Figure 14**Error! Reference source not found.**a) and the debris
slides off the oversteepened, or because locally differential melting steepens the channel walls until the debris slides off. The
location of supraglacial meltwater channels on Zmuttgletscher seems closely related to areas of compressional flow – in flat

and stagnating areas  — and has been most pronounced in the lowest ~1.5 km, where also most  ice cliffs) occur. Th consistent with the cessation in  up-glacier migration of the ice-cliff boundary  in 2005, just below a step in the surface  with extensional flow (Figure 12a at profile distance ~3500 m). Ice cliffs also develop when en- and subglacial  voids  collapse and the shear planes of the fractured ice get exposed as also observed at Zmuttgletscher (Figure 16(Figure 14b). We  suggest that  englacial ablation is effectively creating and enlarging such  voids (Benn et al., 2012; Immerzeel et al., 2014; Fischer 2015; Figure 14**Error! Reference source not found.**c) . Supraglacial ponds are known to have a high potential to increase ablation (Sakai et al., 2000; Röhl, 2008; Miles et al., 2016) and are also potential origins of ice cliffs (Figure 14**Error! Reference source not found.**d) but these features are not prevalent on  Zmuttgletscher .

[Figure]

Large  ice cliffs also exist at the terminus and are responsible for exacerbated retreat  compared to where the terminus is gently sloping and debris-mantled. The situation of a terminal cliff at the glacier mouth combined with terminus retreat is also found e.g. at Gangotri glacier (Bhambri et al., 2012; Bhattacharya et al., 2016), whereas some glaciers with a stable terminus (e.g. Khumbu, Miage) do not show such terminal cliffs (Bolch et al., 2008a; Diolaiuti et al., 2009). Remarkably, depressions and  irregular surface topography near the terminus  were already indicated in the Siegfried map from 1879. Certain  ice cliffs on Zmuttgletscher have reached >25 m in height and have persisted for over two decades. The consequences of ice cliffs at Zmuttgletscher are: (i) a more chaotic pattern of surface elevation changes due strong  local ablation, (ii) a debris redistribution  through cliff backwasting and  fluvial transport, (iii) and stronger surface elevation changes at the lower part of the tongue (especially downstream of  topographic step~~s, see Figure 15a at profile distance ~3500 m, which would not be the casethese exposed-ice areas~~
[revised manuscript text omitted]

---

## Author Response (AR2)

**Reply to reviewer's comments**

N. Mölg et al.

In the following we reply to the specific and technical comments to the revised version of the manuscript.

**Comments Reviewer 1:**

- Are acronyms for "TMG", "SBG", etc. really necessary? Perhaps for figures to save space (and then they should be stated in each caption), but given space is not an issue electronically and these only save several letters, I highly suggest avoiding the use of needless acronyms that only reduce readability by having the reader by having to look them up constantly.

Done. Acronyms have been removed throughout the text.

- Figure 4: There doesn't seem to be any statement of comparing to the other glaciers like there was for Figure 3. Add this statement otherwise the reader is left wondering why all these other glaciers are shown. Zmuttgletscher should be the first in the legend and it should also be more pronounced (perhaps a thicker line, black color, or both). The abbreviations are not provided in the caption. It also seems unnecessary to have these glacier name abbreviations only one time in the text (especially for the study area glacier!). I highly recommend providing the full names, which may be possible if the legend is one column. If you still favor abbreviations, then the abbreviations should be stated in the caption. Additionally, "Zmuttgletscher shows a relatively modest retreat..." - this sentence belongs in the text, not in the caption, and then makes sense as to why you add the other glaciers and address the first part of this comment.

Done. We've moved the sentence from the caption to the text, thus the reference to the comparison should be clear. Also, the figure was changed and Zmuttgletscher is now on top (also in the legend) and the lines are black and thicker. There are no more abbreviations in the legend.

- Figure 8: The labels are now acronyms of acronyms. This is very confusing and a bit ridiculous. Just state SBG1, SBG2, TMG1, TMG2 if you're limited for space in the inset figure.

Done as suggested.

- Figure 9: The plot is very misleading as it shows periods of time that are not equidistant; ex. the first one is 16 years and the second is 6 years, but they have the same space between. I would plot the years and then plot the values at the mid-points as this would show the time more accurately. Also, it'd be nice to be able to compare the timing of these changes with the data in Table 3. I would recommend plotting Table 3 as subfigures to Figure 9. This is why having the actual time on the x-axis would be nice as well. This would allow one to see that in 1983-1988, there was also a decrease in debris-covered glacier area.

Figure has been adapted. The table was left outside the figure according to the journals requirements.

- Supplementary table captions should be above tables like the text

Done.

\- Supplementary figures 7-16 appear to be the same as Figure 10 - delete these in the supplement (also they should each have a proper caption if they were to remain for some reason)

We've kept them in the supplements because this way the interested reader can study them in detail. We'va adapted the captions.

P1, L13 - delete "with decadal"

Done

P1, L16 - "by comparison"

Done

P1, L19 - glacier "has" been quasi-stagnant

Done

P1, L20 - in "the" surface slope

Done

P1, L24 - decreasing glacier dynamics sounds odd. More accurate would be decreasing flux divergence, ice flux, or whatever way the authors would prefer to state this.

Changed to 'decreasing ice flux'

P2, L16 - (iv) "long-term (> decade) glacier-scale studies have mostly..."

Done

P2, L28 - add reference for long response times (>50 years)

Done

P3, L8 - weather "systems"

Done

P3, L9 - change units to m in order to be consistent with the next line of m w.e.

Done

P3, L14 - Zmuttgletscher "has" several independent...

Done

P3, L20 - "mainly fed by TMG and to a lesser extent..."

Done

P4, L6 - our analysis "is" based

Done

P5, L4 - new paragraph seems unnecessary

Done

P5, L8 - using "the Topo-to-Raster tool in ArcGIS."

Done

P5, L13 - "and then georeferenced"
  Done

P5, L17 - "the number of images and image quality..."
  Done

P5, L28 - ... "(1859), Siegfried map (1879), all available orthophotos, and Swisstop (...) by manual digitization"
  Done

P5, L29 - Swisstopo images? maps? stating simply "Swisstopo" sounds odd
  Done

P6, L1 - "The time series of maps and orthophotos resulted in glacier area values ..."
  Done

P6, L2 - "debris-covered areas" compared to...
  Done

P6, L2 - of "these" images
  Done

P6, L12 - influence "of" debris cover
  Done

P6, L14 - glacier boundary "and" the start and end of the length profile", respectively ..."
Figure 2 - state in the caption that c & d are looking at Tiefmattengletscher to provide the reader with the proper context for looking at the images
  Done

P7, L2 - the "long-term elevation change data"
  Done

P7, L9 - delete "results of upper transect see Figure 7a" as this is unnecessary
  Done

P7, L11 - "However, for debris thicknesses greater than" 20 cm...
  Done

P7, L14 - add the time periods used by each station in the parentheses since you provide the elevation and distance from the site. I would also switch the order to be consistent with the text (i.e., close to study site, similar elevation, long period)
  Done

P7, L20 - into four "sections" and eleven...
  Done

P7, L26 - Between "August 22-24 2017"...
  Done

P7, L27 - replace "today's" terminus with the year. Many years from now "today's" terminus will not be accurate. I would suggest this for each use of "today" in the text
  Done

P8, L10 - specify "9b"
  Done

P8, L15 - A glacier's elevation change gradient is typically inclined towards ... - this sentence is very confusing to read. Please clarify. The gradient is a number, so inclining a number doesn't make sense.
  Done

P8, L35 - accounted for "due to a" lack of such data
  Done

P9, L6 - delete "negative at"
  Done

P9, L8 - Zmuttgletscher "had less" mass loss
  Done

P9, L8 - add citation for Swiss mean
  Done

P10, L14 - delete ~12.9% of the entire glacier area, and move this to the next line, so it reads "resulting in an increase in percent debris cover from ~12.9% to 31.8+/-0.06% of the glacier area from ~1850 to 2013." Also, add error for the 12.9% if able to do so, since you show it for the 2013 image.
  Done

P10, L21 - and "now is" close to the foot...
  Done

P11, L8 - from "less than" 5 cm to "greater than" 70 cm
  Done

P11, L16 - seven "week" period
  Done

P11, L17 - if stating "Ostrem-like" behavior then should add reference to the Ostrem study here
  Done

P12, L6 - The "middle" section ... - this is then consistent with term middle in Figure 8
  Done

P12, L20 - delete "and more"
  Done

P13, L21 - The use of "on average" with all the years sounds like it's setting up an average for the time period, only to realize that it's meant for spatial average; hence, I would recommend

something along the lines of "Since 1879, the average thinning over the ablation area has been 104.7...". Although even here, I would suggest putting the value in m/yr because 104 m is a lot but it lacks the context to be able to compare it to all the other thinning rates mentioned in this paragraph.

Done

P15, L7 - delete "rather"

Done

P15, L18 - which "provide" a clear picture

Done

P15, L30 - delete "in our case"

Done

P16, L31 - develop "a" more negative mass balance

Done

P17, L9 - delete "points"

We are still convinced that the 'points' are actually correct. Removing them would leave the reader confused as to whether the change is in % relative to the former area or absolute % change.

P18, L10 - Table S15 doesn't exist

Changed

P21, L28 - Recommend changing Owen to "O." to be consistent with the other acknowledgements

Done.

**Comments reviewer 2:**

P1 L13: There is something missing near 'with decadal …'
  Done

P3 L10: 15 km distant -> distance
  Left as is. Distant should be correct.

P4 L6: Our analyses are based
  Done

P6 L3: West -> west
  Done

P7 L25: Please also mention the actual visually determined threshold here.
  Done.

P7 L25: Not very clear from the pdf whether there is a split, but consider splitting paragraph after 'water surfaces' ?
  Done

P11 F6: Nice!
  Thanks!

P13 L26: Relationshipt -> relationship -> although I would prefer just 'relation'.
  Done

P13 L27: …show a trend to stronger thinning… Vague to me, can this be rephrased?
  Done

P15 L14: investigation priod -> study period
  Done

P16 L32: 0.8-1 -> 0.8-1.0
  Done

P17 S5.3. I am still bit baffled about the need for percentage points, but OK.
  Left as is.

P17 L11: Paper by Teun van Woerkom is no longer in discussions and has been published now. https://doi.org/10.5194/esurf-7-411-2019
  Included

**Comments Editor**

The contribution of 1.5 days of radar interferometry measurement is so negligible to your story that it could be skipped. To really focus the paper and not district the reader.

   We are of the opinion that this piece of information should be kept in the paper, since it is the only distributed information about glacier velocity, even from three points in time.

Technical corrections.
Table 1. Dh not defined yet (I think)
   Done

5.23. Maybe remind here the date of the reference DEM to which all others are coregistered.
   Done

7.19 too large rather than too big. (I think).
   Done

8.30 Does not really make sense to have a section 3.6.1 if you do not have 3.6.2.
   Done

Table 2. Even if you write "annual" I think it is good for clarity that the unit in the table is set to m/yr. To avoid any mis-understanding.
   Done

Do your glacier-wide mass balance agrees with the one from Fischer et al., TC, 2015 when cumulated over a similar period? It would be a nice sanity check.
   Yes it does: -0.54 (ours from 77-2010) vs. -0.65 (theirs, 80-2010)

Figure 7. The last sentence of the caption appears to be incomplete.
   Done

16.32. Minus sign missing in front of the mass balance.
   Done

18.10. Not clear why you say "comparatively high" here and then provide the example of Findeleng. where the area loss is even larger.
   Changed to 'low'

19.14. The same statement applies I think to Argentière Glacier in the French Alps (see Vincent, C., Soruco, A., Six, D. and Le Meur, E.: Glacier thickening and decay analysis from 50 years of glaciological observations performed on Glacier d'Argentière, Mont Blanc area, France, Ann. Glaciol., 50(50), 73–79, doi:10.3189/172756409787769500, 2009.)
   Included

20.19. I had a hard time tracking back the 5% contribution of ice cliff to the glacier-wide mass balance. Better explain how you calculated this 5% value (if not already done)
   Done

20.25 and 20.29 are mostly repetitions.
   Left as is